

# Potential of optical and ecological proxies to quantify phytoplankton carbon in oligotrophic waters

David Antoine[1,2], Chandanlal Parida[1], Camille Grimaldi[3]

[1] Remote Sensing and Satellite Research Group, School of Earth & Planetary Sciences, Curtin University, Perth, Australia

[2] Sorbonne Université, CNRS, Laboratoire d'Océanographie de Villefranche, Villefranche sur mer 06230, France

[3] University of Western Australia, Indian Ocean Marine Research Centre, Fairway, Crawley, WA 6009

*Correspondence to*: Chandanlal Parida (chandan.parida@curtin.edu.au)





## Abstract

Satellite ocean color observations provide two proxies to estimate the phytoplankton carbon concentration, $C_{phyto}$, then used as input to models quantifying growth rates and primary production, namely the phytoplankton chlorophyll-a concentration, Chl-a, and the particulate backscattering coefficient, $b_{bp}$. Variability in phytoplankton community composition, pigment assemblages and contribution of non-algal material all interplay in the relation between these proxies and $C_{phyto}$, so that no ubiquitous relationship exists

between them. It is accordingly still unclear which of Chl-a or $b_{bp}$ is best suited to quantify $C_{phyto}$, or whether they both are yet each in specific trophic conditions, especially for low-productivity oligotrophic waters. Here we use a data set from the eastern Indian Ocean that includes phytoplankton cell counts, phytoplankton pigments, particulate organic carbon (POC) and inherent optical properties (IOPs) to perform a comparative assessment of $C_{phyto}$ derived from either Chl-a or $b_{bp}$ or cell counts combined with allometric relationships.

We found significant correlations ($r^2 > \sim0.5\text{-}0.6$) between the three $C_{phyto}$ estimates and IOPs, Chl-a or POC when samples from all depths down to 150 m are included. When only the top 25 m are included (amenable to ocean color remote sensing), no significant relationships were found, except between the cytometry-derived $C_{phyto}$ and both Chl and POC. The $b_{bp}$-derived $C_{phyto}$ showed the smallest variability across the entire data set. These results warn about applying to satellite ocean color observations relationships derived from

data collected throughout the euphotic layer.



## 1. Introduction

Phytoplankton are primary producers in marine ecosystems and play a central role in the global carbon cycle. They generate about 45 Giga tons of organic matter every year in the global ocean (Field et al., 1998; Falkowski, 2012). A fraction of this organic matter sinks into the deep ocean, contributing to lowering the $CO_2$ partial pressure of surface waters. This process reduces atmospheric $CO_2$ levels by ~200 ppm as compared to a theoretical abiotic ocean (Parekh et al., 2006). Given their role in carbon sequestration, accurately quantifying phytoplankton carbon biomass and productivity is essential.

Oligotrophic regions, where phytoplankton growth is limited by low nutrient levels and their biomass and chlorophyll concentration accordingly low, represent a major uncertainty in the global phytoplankton productivity budget.

Satellite ocean color observations, from which phytoplankton chlorophyll-a (Chl-a) can be derived, have allowed estimates of phytoplankton productivity at global scale. Chlorophyll is not an accurate measure of phytoplankton biomass, however, as phytoplankton can adjust their level of chlorophyll to adapt to changing nutrient and light conditions without necessarily showing concomitant changes in their carbon content ($C_{phyto}$), through a process referred to as photo-acclimation. They also contain accessory pigments whose ratios to Chl-a vary with species composition and environmental conditions, complicating biomass quantification (Cloern et al., 1995).

It has been suggested that the particulate backscattering coefficient, $b_{bp}$, derivable from satellite ocean color observations in parallel to Chl-a, is a relevant proxy of $C_{phyto}$ in the marine environment (Behrenfeld et al., 2005; Graff et al., 2015). Models have been developed based on this concept (Westberry et al., 2008; Silsbe et al., 2016) and compared with Chl-based or absorption-based models, showing some significant differences in the distribution and global amounts of phytoplankton primary production (Westberry et al., 2023). Deriving $C_{phyto}$ from the analysis of the relationship between chlorophyll concentration and the concentration of particulate organic carbon (POC) has also been proposed (Sathyendranath et al., 2009). As far as field measurements are concerned, using cell counts from cytometry analyses combined with allometric relationships has been used to derive $C_{phyto}$ and analyze its relation to $b_{bp}$ (e.g., Martinez-Vicente et al. (2013) ). Correlation was confirmed yet a variety of slopes are observed when a linear relationship between both is derived from datasets covering various environments (Serra-Pompei et al., 2023; Qiu et al., 2021). Direct measurements of $C_{phyto}$ in natural samples are difficult to carry out, however, so none of these approaches have yet been thoroughly validated. To the best of our knowledge, Graff et al. (2015) remains the only study to date that has used direct measurements of $C_{phyto}$ in relation to $b_{bp}$.

Studies on $C_{phyto}$ remain relatively scarce and estimating $C_{phyto}$ in oligotrophic waters remains a significant methodological challenge in oceanographic research. Each method has its own advantages and



limitations, often relying on simplified empirical representations of phytoplankton's physiological variability. However, these methods tend to perform poorly when applied to different oceanographic regions where conditions change significantly. Phytoplankton acclimate to variations in light, nutrients, and temperature by modulating cellular pigment levels to meet the altered demands for photosynthesis. This acclimation is

effectively characterized by shifts in the ratio of chlorophyll to carbon biomass (Macintyre et al., 2002; Sakshaug et al., 1989).

The Eastern Indian Ocean (EIO) offers food, natural resources and numerous benefits to surrounding countries (Hermes et al., 2019). In the northern EIO, the Bay of Bengal experiences monsoon-driven seasonal circulation changes during summer (southwest monsoon) and winter (northeast monsoon) (Schott and

Mccreary Jr, 2001), along with large freshwater inputs from the rivers and rainfall that create surface stratification and barrier layer (Vinayachandran, 2009). At the equator, strong westerly winds during spring and fall generate the Wyrtki jet (Wyrtki, 1973). In the south EIO, the westward-flowing Southern Equatorial Current (SEC) carries low-salinity water, separating the south Indian Subtropical Gyre from the northern EIO.

The oligotrophic southern EIO is less productive than other regions of the Indian Ocean. It has remained relatively unexplored regarding the distribution of phytoplankton carbon, especially compared to the Pacific and Atlantic Oceans. During the Second International Indian Ocean Expedition (IIOE-2) (Hood et al., 2015), the 110° E line was revisited in May/June 2019 by R/V Investigator to replicate as closely as possible the sampling stations of voyages of the HMAS Diamantina in May-June 1963. In addition to revisiting the 1963

survey stations to evaluate 60 years of change in the southern EIO, new parameters such as bio-optical measurements, pigment concentrations, and cell abundance were measured for the first time.

The 110° E line cruise provided a unique opportunity to explore different ways of deriving $C_{phyto}$ in oligotrophic waters either from the relationship between the POC and Chl-a concentrations or from the optical backscattering coefficient (an optical proxy) or from detailed information of the phytoplankton population

from cytometry measurements. The impact of the phytoplankton community composition, inferred from pigments, on the results obtained through these three methods was assessed and their limitations discussed.

## 2. Data and methods

The field data used in this study were acquired along the 110° E meridian in the southeastern Indian Ocean from 17 May to 5 June 2019 on the Australian Marine National Facility, R/V Investigator. This research

voyage (IN2019_V03) was part of the International Indian Ocean Expedition (IIOE-2). A total of 20 stations were occupied from 39.5° (station 1) to 11.5° S (station 20), each ~1.5° apart (Fig. 1). Conductivity-temperature-depth (CTD) casts were undertaken at each station using a Sea-Bird Scientific (Bellevue, Washington), SBE911. At each station, a first cast was going deep down to 10 m above the seafloor and was





conducted around 7 am Australian Western Standard Time. A second cast was shallower (~500 m) and

carried out around 8 pm. At each CTD cast, water samples were collected from various depths from surface

to 400 m using a 36 12L-Niskin-bottle rosette. The water was then used for onboard cytometry analyses or

filtered for subsequent post-cruise laboratory analyses (see below). Optical instruments were operated

immediately after the morning CTD cast to collect collocated data.

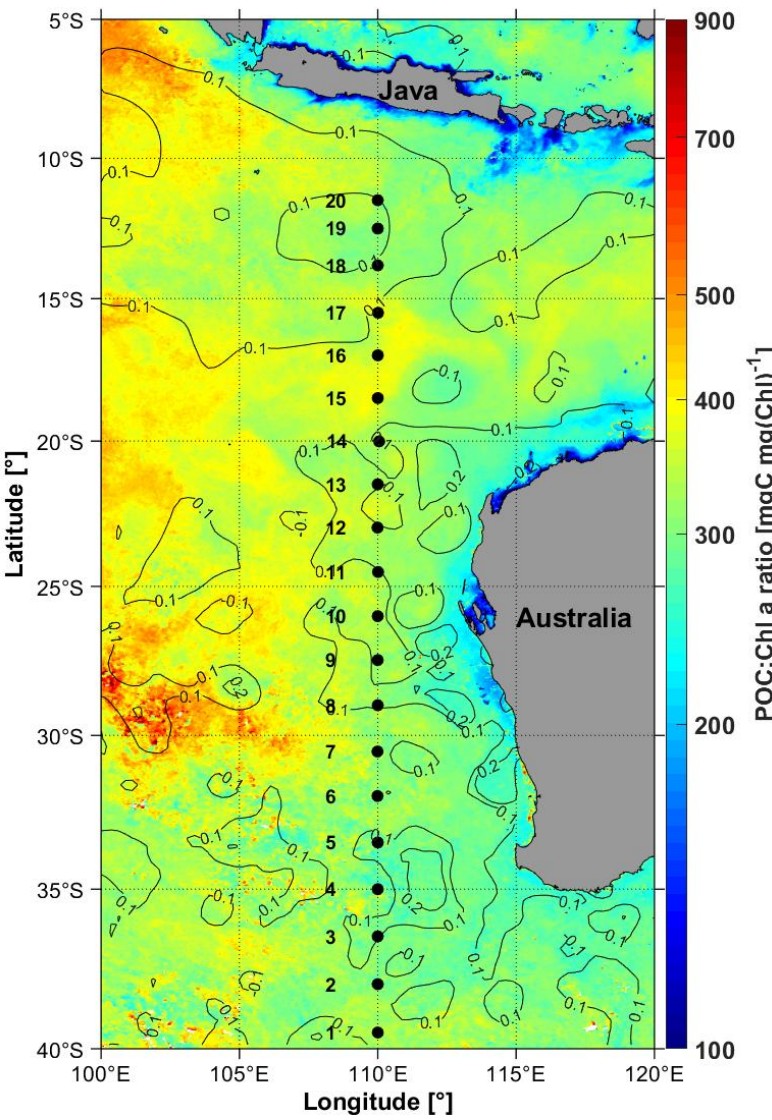

**Fig. 1.** Location of the 20 sampling stations visited during the 110 °E voyage (black dots) from 17[th] May

(station 1) to 5[th] June 2019 (station 20) overlayed with the monthly mean ratio of particulate organic carbon

(POC) to chlorophyll-a (Chl-a) for the month of May 2019, derived from MODIS Aqua. The overlaid

contours are averaged Sea Level Anomalies (SLA, m) from the Jason-3 altimeter mission.



## 2.1. Phytoplankton pigments

Typically, 2.3 L water samples were filtered through 25 mm diameter Whatman glass fiber filters (GF/F 0.7 µm particle retention size) under low vacuum pressure. Then the filters were placed in petri dishes wrapped with aluminum foil, flash-frozen in liquid nitrogen, and then stored onboard at -80 °C. After the cruise, the samples were stored back into a liquid nitrogen dry shipper for being sent for analysis by the "*Service d'Analyse de pigments par HPLC*" (SAPIGH) of the *Institut de la Mer de Villefranche*, France, where they were also stored at -80°C until being analyzed. Phytoplankton pigment concentrations were then determined through High Precision Liquid Chromatography (HPLC) using an Agilent Technologies 1200 Series equipped with an Eclipse XDB C8 column, following the method by Ras et al. (2008). Pigments were extracted for 2 hours in 3 ml 100% methanol, disrupted by sonication, and clarified by vacuum filtration through Whatman GF/F. Further details can be found in Parida and Antoine (2025).

The total chlorophyll a (TChl-a) is here defined as the sum of monovinyl-chlorophyll-a and divinyl-chlorophyll-a. The dominance of either photosynthetic carotenoids (PSC) or photoprotective carotenoids (PPC) is used as an indicator of the phytoplankton population status. Included here in the PSC are the 19-Hexanoyloxyfucoxanthin (19-Hex), 19-Butanoyloxyfucoxanthin (19-But), Fucoxanthin (Fuco), and Peridinin (Peri), while the PPC are Alloxanthin (Allo), Total carotene (Tcar), Diadinoxanthin (Diadino), and Zeaxanthin (Zea). The PPC-dominated waters here correspond to depths < ~50 m at the most oligotrophic stations of the transect.

## 2.2. Particulate organic carbon (POC)

Water samples for particulate organic carbon (POC) were collected during the morning CTD upcasts only. Three-liter samples were collected from Niskin bottles directly into pre-washed polycarbonate carboys. Water was filtered on pre-combusted (450 °C for 4hr) 25-mm Whatman GF/F filters at low vacuum. Filters were stored in acid-washed Petri dishes, dried in an incubator at 55 °C for 24-48h then stored in a desiccator with concentration HCL fumes for 24h to remove inorganic carbonate. The filters were then dried again at 55 °C for 48 hours before being folded and packed into pre-combusted tin capsules and stored at -20° C until laboratory analysis.

Samples were analyzed by the Research Corporation of the University of Hawaii, Hl, on a Costech ECS 4010 Elemental Combustion System using a Zero Blank Autosampler.

Finaly, POC concentrations were calculated by subtracting the average concentration of carbon in dry blank filters. The average value of these blanks was $14.3 \pm 3.1$ mg m$^{-3}$.

## 2.3. Cell counts



Flow cytometric analyses and cell sorting were performed using a BD Influx™ cell sorter (BD Biosciences) located in a laboratory onboard the Research Vessel Investigator (Commonwealth Scientific and Industrial Research Organisation). This instrument was equipped with multiple lasers and photomultiplier tubes for optical signal collection, including an optical pulse shape signals from a 488 nm blue laser, along with forward scatter (FSC), side scatter (SSC) and fluorescence detection in the orange

(580/30 nm) and red (692/40 nm) channels.

Prior to each run, the instrument and workspace were prepared following a standardized protocol to ensure sterility, fluidic stability, and accurate optical alignment. Before operation, the cytometer was powered on following the manufacturer's startup sequence, which included powering the electronics board, operator computer, and activating the instrument software (BD FACS™ Software). A wet start was performed to

remove air bubbles and clean the sample lines. The sample line was flushed sequentially with 10% bleach (5 minutes) and MilliQ water (5 minutes), with intermediate backflush steps.

Instrument quality control (QC) and alignment were performed prior to data acquisition. Stream alignment was adjusted using in-built video feedback and fine-tuning knobs to centre the fluid stream within the pinhole. Optical alignment was conducted using Single Peak and Eight Peak fluorescent calibration beads.

Bead samples were run in "Sample Boost" mode, and laser alignment was optimized to achieve tight and bright bead populations in forward scatter and fluorescence plots across blue, red, and UV lasers.

Once calibrated, the samples (with initial sample volume of ~ 8 mL) were run for counting to achieve 1.8-2 mL of final sorted sample. Where needed, samples were pre-filtered using 100 μm Nitex mesh to prevent clogging of the nozzle tip. Acquisition settings were set to record up to 1,000,000 events or for a

fixed duration of 120 seconds. Sample data were recorded through the BD FACS™ software.  Samples were run in a default daily workspace and collected under defined pressure settings with regular monitoring of sheath and sample pressures to maintain consistency across runs. Immediately after sorting, samples were mixed gently with a pipette, and a 100–150 μL aliquot was used to perform a post-sort recount. This recount measured both total cell number and runtime to assess sample concentration. A volume calibration was also

conducted by running 1 mL of MilliQ water through the cytometer for a fixed time of 2 minutes. The post-run volume was measured to determine the effective flow rate and correct for sample recovery. Both the sorted cell sample and the filtered sheath fluid were snap-frozen in liquid nitrogen and stored at –80 °C for downstream analyses. After each sort, sorting chambers and flow paths were cleaned using bleach and MilliQ water, and sample lines were flushed to remove residual beads or cells.

After acquisition, the data from each event was visualized using Forward Scatter and Side Scatter to identify and isolate specific populations of interest within the mixture of cells based on their physical and/or fluorescent characteristics. A first gate was typically drawn to exclude debris (very low FSC and SSC). Based on fluorescence markers or scatter properties, gates were applied to define and quantify specific



subpopulations, including Prochlorococcus, Synechococcus and Picoeukaryotes.

**2.4. Inherent optical properties (IOPs)**

A 25-cm pathlength Western Environmental Technology Laboratories (WET Labs) Inc. (Philomath, Oregon) C-star transmissometer was connected to the auxiliary channels of the CTD to measure the transmittance (Tr, %) at 660 nm at each cast. Prior to CTD casts, instrument windows were cleaned with tissue paper and isopropyl alcohol. The absorption of colored dissolved organic matter at 660 nm is assumed
to be negligible, so the particulate beam attenuation coefficient, $c_p$, is obtained as follows:

$$c_p(660) = \frac{-\ln(Tr)}{0.25}, \qquad [\text{m}^{-1}] \tag{1}$$

where Tr is the transmission corrected for the water contribution and 0.25 the path length in meter.

A Hydro-Optics, Biology and Instrumentation Laboratories (HOBILabs Inc., San Diego, California) Hydroscat-6 (Maffione and Dana, 1997) was deployed to measure the total volume scattering function, $\beta$,
($\text{m}^{-1}$ $\text{sr}^{-1}$) at 140 degrees and six wavelengths (420, 442, 470, 510, 590, 700 nm). This instrument was part of an optical package deployed at each station immediately after the CTD cast. The sensor was factory calibrated before and after the cruise and dark cast measurements were performed in situ systematically before each cast, using black electric tape covering the instrument windows. The deployment included a rinse and temperature equilibration cast down to 50 m and back to the surface, after which the full cast started down to
~ 200 m at a descending speed of ~0.2 m $\text{s}^{-1}$. Only the measurements taken during the downcast were used for this study, as particles could be significantly disturbed by the wave generated by the package itself during the upcast. Each cast was either preceded or followed by a dark cast for which the instrument emitting and receiving windows were masked using black electric tape. The particulate backscattering ($b_{bp}$) was then derived from $\beta(140°)$ as follows (Maffione and Dana, 1997):

$$b_{bp}(\lambda) = 2\pi\chi\big[[\beta(140°,\lambda) - \beta_{dark}(140°,\lambda)] - \beta_w(140°,\lambda)\big], \quad [\text{m}^{-1}] \tag{2}$$

where $\beta_{dark}(140°, \lambda)$ is the average dark value from all dark casts. The average value was used because all dark values were close to 1.5 $10^{-6}$ $\text{m}^{-1}$ $\text{sr}^{-1}$ with standard deviation of ~2.5 $10^{-7}$ $\text{m}^{-1}$ $\text{sr}^{-1}$. The $p$ and $w$ subscripts represent the scattering contribution from the particles and seawater respectively, and $\chi$ is the conversion factor between $b_{bp}$ and the particle volume scattering function, here set at 1.0807 from the instrument
calibration. The water volume scattering function $\beta_w(140)$ is computed from Zhang et al. (2009) using temperature and salinity from a Seabird SBE49 FastCAT CTD deployed on the optical package.

The $b_{bp}$ wavelength dependence is assumed to follow a power function:

$$b_{bp}(\lambda) = b_{bp}(\lambda_0)\left(\frac{\lambda_0}{\lambda}\right)^{\eta}, \qquad [\text{m}^{-1}] \tag{3}$$




where $\lambda_0$ is a reference wavelength and $\eta$ is the dimensionless spectral slope. A linear least-squares fit on the
log-transformed $b_{bp}$ and wavelength ratios gave $\eta = 1.63$. This slope is used to convert our $b_{bp}$ values at 470
nm to the appropriate wavelength before introducing them into equations from previous studies that used
other wavelengths than 470 nm. The band at 470 nm was actually standing out of the fit to the six spectral
bands, requiring a +20% correction to align it with the spectral slope (see Supporting Information Fig. S1).
This lower backscattering coefficient cannot be explained by the anomalous dispersion effect because
absorption at this wavelength in our data set is very low. It is therefore likely due to a bias in calibration at
this wavelength.

Finally, a 7-point running median filter as described by Briggs et al. (2011) was applied to the $b_{bp}$
vertical profiles before 5-m average values were calculated for each of the rosette sampling depths ±2.5 m.

### 2.5. Phytoplankton carbon

We tested three different and independent methods to calculate $C_{phyto}$, where it is estimated either through
the relationship between Chl and POC, from the particulate backscattering coefficient, $b_{bp}$, or from the cell
abundances measured by cytometry and their cellular carbon. These methods are briefly described below.

The POC-based method was suggested by Sathyendranath et al. (2009). It assumes that for any given
chlorophyll concentration, the lowest POC content represents the upper bound on $C_{phyto}$. The $C_{phyto}$ vs. TChl-
a relationship is obtained through a 1% quantile regression applied to log-transformed POC and TChl-a data
derived from the CTD rosette water sampling.

The second method was initially suggested by Behrenfeld et al. (2005). It uses the particulate
backscattering coefficient, $b_{bp}$, the underlying idea being to avoid confounding Chl-a changes that are only
due to photoacclimation, i.e., Chl-a changes not accompanied by changes in carbon content, with Chl-a
changes concomitant to changes in $C_{phy}$. We used the Graff et al. (2015) $C_{phyto}$ vs. $b_{bp}$ (470) relationship,
which is to the best of our knowledge the only such relationship based on $C_{phyto}$ values directly measured
across diverse marine environments, ranging from the oligotrophic gyres of the pacific and Atlantic to
equatorial upwelling regions. Their relationship is:

$$C_{phyto} = 12,128 \times b_{bp}\,(470) + 0.59 \quad [\text{mgC m}^{-3}] \tag{4}$$

The third method uses cytometry-derived phytoplankton cell counts and estimates of biovolume
conversion factors to calculate their cellular carbon content, similarly to what Martinez-Vicente et al. (2013)
did, and here following Qiu et al. (2021):

$$C_{pico} = \sum_{i=1}^{3} 10^{-6} \cdot N_i \cdot \epsilon_i \cdot \left(\frac{\pi}{6}\, D_i^3\right), \qquad [\text{mgC m}^{-3}] \tag{5}$$

Where $C_{pico}$ is the phytoplankton carbon for pico-phytoplankton only. The three groups considered (*i* from 1



to 3) are *Prochlorococcus, Synechococcus,* and picoeukaryotes. In Eq. (5), $N_i$ is the cell abundance (cell mL$^{-1}$), $\varepsilon_i$ is the bio-volume conversion factor (fg C $\mu m^{-3}$), $D_i$ is the cell diameter ($\mu$m), and $10^{-6}$ is the unit conversion from fg C mL$^{-1}$ to mg C m$^{-3}$. We used the average cell diameters ($D_i$) proposed by Martinez-Vicente et al. (2013) (Their Table 1), i.e., 0.68 $\mu$m for *Prochlorococcus*, 1.22 $\mu$m for *Synechococcus* and 1.56 $\mu$m for picoeukaryotes. The $C_{phyto}$ *vs* $b_{bp}$ relationship that they derived has a large negative slope,

however, suggesting that the conversion factors they used might have been too low. We therefore rather used $\varepsilon = 280$ fg C $\mu m^{-3}$ for *Prochlorococcus* and *Synechococcus*, following Heldal et al. (2003). The range of values reported for the conversion factor for pico-eukaryotic algae seems quite large. We used a value of 380 fg C $\mu m^{-3}$ from Garrison et al. (2000) and we also tested a value of 750 fg C $\mu m^{-3}$, as derived from a cellular carbon of 1500 fg per cell (Zubkov et al., 1998) and the cell diameter of 1.56 $\mu$m.

Because flow cytometry typically only detects cells smaller than 3 $\mu$m, larger cells are omitted from the analysis. Therefore, the total phytoplankton carbon biomass ($C_{phyto}$) was estimated from $C_{pico}$ using the empirical ratio $f_{fc}$ (Roy et al., 2017):

$$f_{fc} = \frac{3^{3q-\xi+1}-0.2^{3q-\xi+1}}{200^{3q-\xi+1}-0.2^{3q-\xi+1}}, \qquad \text{[dless]} \qquad (6)$$

where 200 is the upper boundary of phytoplankton sizes in micrometers. The allometric parameter q was set

to 0.85 (Roy et al., 2017). In Eq. (6), the slope of the phytoplankton particle size distribution, $\xi$, was derived from the phytoplankton chlorophyll-specific absorption coefficient at 676 nm, $a^*_{ph}(676)$, as also suggested by Roy et al. (2017). We approximated their relationship by linearly interpolating between values of $\xi = 3.5$ when $a^*_{ph}(676) = 0.015$ m$^2$ mg$^{-1}$ and of $\xi = 4.5$ when $a^*_{ph}(676) = 0.035$ m$^2$ mg$^{-1}$ (see their Fig. 1a). The phytoplankton absorption was derived from filter-pad measurements as described in Parida and Antoine

250 (2025).

    $C_{phyto}$ is finally obtained as:

$$C_{phyto} = \frac{c_{pico}}{f_{fc}} \qquad \text{[mgC m}^{-3}\text{]} \qquad (7)$$

    Our $f_{fc}$ values ranged from 0.6 to 0.92 (0.778±0.09, average ± standard deviation), which is consistent with previous studies and confirms that pico-phytoplankton make a substantial contribution to the total

phytoplankton carbon biomass in oligotrophic waters as those we encountered here.

## 2.6. Data uncertainties

    Uncertainties assigned to field measurements are generally of Type B (guide to the expression of uncertainty in measurement, GUM2010 (Bipm et al., 2008)). Type A uncertainties require repeated measurements of the same quantity in controlled conditions, which is not what field measurements generally





allow. Standard measurement and analysis protocols were followed so we have taken the approach to rely on generally agreed measurement uncertainties for most parameters, except when we had other independent measurements or information allowing us to be more specific.

For TChl-a, we used a 15% uncertainty as generally observed from intercomparison exercises involving multiple HPLC analysis facilities (e.g., Hooker et al. (2012)). For POC, we assumed that the standard
deviation of measurements made on retaining-particle filters is the same as that we quantified from the blank filters ($\pm 3.1$ mg m$^{-3}$) and therefore we used a quadratic error to combine uncertainties from the blanks and particle-retaining filters into a value of $\pm 4.4$ mg m$^{-3}$. This is close to what Sandoval et al. (2022) found through a detailed analysis of all potential uncertainty sources in experimental determination of POC (about 2.5 mg m$^{-3}$).

Uncertainty for $b_{bp}(470)$ was assessed by comparing values derived from simultaneous measurements by the Hydroscat-6 instrument used here and by another optical backscattering meter, the In-Situ Marine Optics Pty Ltd (Perth, Australia) SC6. The parallel deployment was carried out during a research voyage in the Southern Ocean in waters with chlorophyll concentrations from 0.1 to 0.5 mg m$^{-3}$. The comparison showed an average root mean square error (RMSE) of 2 10$^{-4}$ m$^{-1}$ for $b_{bp}(470)$. For $c_p(660)$, a constant value of 0.015
m$^{-1}$ was used, corresponding to the standard deviation calculated on deep values (depth > 500 m) for the 20 casts (assuming deep values should be constant).

For the $C_{phyto}$ derived from the cytometry data, we used a Monte Carlo approach as Martinez-Vicente et al. (2013) did, leading to an estimate closer to a type A uncertainty. The procedure consisted in running Eqs. (5 to 7) 20,000 times for each sample, each time with a different set of the parameters D, $\varepsilon$ and $\xi$ (so $f_{fc}$). For
D and $\varepsilon$, we multiply the standard errors provided in Table 1 of Martinez-Vicente et al. (2013) by a number randomly picked in the [-0.5 – 0.5] interval. Random numbers in the same range multiply an uncertainty set to 6 10$^{-3}$ m$^2$ mg$^{-1}$ for $a^*_{ph}(676)$, the result being then used to calculate the uncertainty of $\xi$ and then of $f_{fc}$. Finally, the square root of the number of cells for each group is used as a measure of dispersion and is, again, multiplied by a random number before being used in Eq. (5). The final uncertainty of each $C_{phyto}$ calculation
is taken as the standard deviation of the resulting 20,000 $C_{phyto}$ values.

As for $C_{phyto}$ derived from either TChl-a or $b_{bp}(470)$, their uncertainty is calculated by introducing the TChl-a or $b_{bp}(470)$ uncertainty mentioned above into the relevant equations.

These uncertainties are used to display error bars on Figures 4 to 8. They are not considered in the regression analyses, however. These analyses are Type II regression performed using the lmodel2 function
in R (Legendre, 2014).



## 3. Results

### 3.1. General traits of the Chl-a, POC, $c_p$ and $b_{bp}$ data sets

The distribution of the monthly mean POC: Chl-a ratio for May 2019 is displayed in Fig. 1 (data from the Aqua-MODIS ocean color sensor). It shows values from about 150 in coastal areas to about 800 mgC

mg(Chl)$^{-1}$ in offshore waters, and from about 300 to 450 mgC mg(Chl)$^{-1}$ for the 20 stations we occupied along the 110-east line. The higher POC:Chl-a ratios in the western part of the area are mostly driven by decreasing Chl-a concentrations when POC concentrations remain relatively stable. The field data for depths < 40 m show the same maximum values yet also include lower ratios, down to 150 mgC mg(Chl)$^{-1}$ in the southern part of the transect (stations 1 to 7) and at stations 19 and 20.

The latitudinal variations in the depth profile of Chl-a (Fig. 2a) show transition from mesotrophic conditions at stations 1-4 (Chl-a around 0.2-0.5 mg m$^{-3}$) to oligotrophic conditions further north from station 5, with concentrations lower than 0.1 mg m$^{-3}$ at surface. A deep chlorophyll maximum (DCM) is observed from station 9 to the end of the transect with a maximum chlorophyll concentration of 0.6 mg m$^{-3}$ at station 15 at a depth of 67 m. At other stations, the DCM is deeper, at about 80 m. Additionally, surface chlorophyll

shows a slight increase at station 18 due to interactions between nearby warm- and cold-core eddies (Phillips et al., 2022). The underwater light field changes as a function of the chlorophyll concentration and cloud cover. During the voyage, the sky was cloudy to overcast at stations 1-6, then cleared up until station 20. Surface downward irradiance between 100 and 450 W m$^{-2}$ were measured along the transect. The euphotic layer depth, determined from vertical profiles of the photosynthetically available radiation (PAR), varied between

60 m and 130 m (red stars in Fig. 2a). This depth was also estimated from the surface chlorophyll concentrations using the Morel and Berthon (1989) model  (magenta line in Fig. 2a). The model predictions align well with the PAR-derived values at all stations but station 13, where discrepancies likely arise from bio-optical properties deviating from model assumptions. Despite this, the model was applied to estimate the euphotic depth for stations 3 and 4, where underwater PAR measurements could not be conducted due to the

weather conditions.



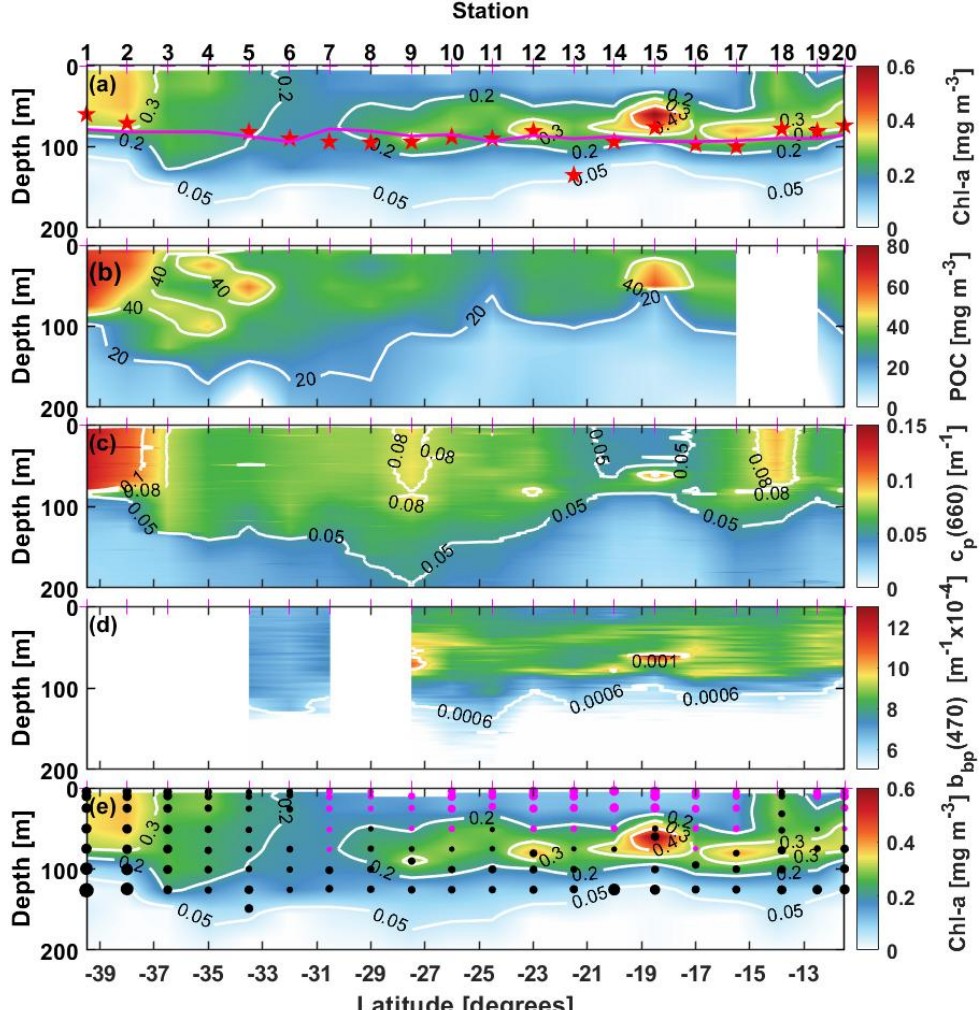

**Fig. 2.** Depth-latitude, colored sections of **(a)** Total Chlorophyll-*a* concentration (TChl-a), **(b)** Particulate organic carbon concentration (POC) **(c)** Particulate beam attenuation coefficient at 660 nm, $c_p$(660), **(d)** Particulate backscattering coefficient at 470 nm, $b_{bp}$(470) and, **(e)** the TChl-a of panel **(a)** with PPC-dominated data points superimposed as pink dots and PSC-dominated points as black dots. The larger the dots the further the ratio is above 1. The station numbers are indicated on top of panel **(a)**. White boxes are missing data.

The POC distribution (Fig. 2b) generally follows that of TChl-a, although POC values in the top 100 m are only twice as high in the mesotrophic stations than at the oligotrophic ones (about 70 vs. 30 mg m$^{-3}$), when a factor of nearly 6 is observed for TChl-a (from 0.08 to 0.5 mg m$^{-3}$). This sems to indicate the presence of a significant pool of non-living organic matter in the oligotrophic part of the transect (elevated POC:TChl-a ratio).



The distribution of $c_p(660)$ (Fig. 2c) follows the general meso- to oligotrophic pattern as POC and TChl-a, with minimum surface values (<0.05 m$^{-1}$) around stations 14-15, where TChl-a is the lowest. It displays three relative maxima, however, which is not what is observed for POC and TChl-a. The largest one, at station 1-3, and the second largest at station 18 are expected because of the larger TChl-a concentrations (eddy-induced at station 18). The third and lower relative maximum around station 9 would not be inferred from the TChl-a distribution, however, which rather shows minimal values at that station. The POC distribution, although not showing minimal values at that station, neither shows a relative maximum there. These larger $c_p(660)$ values seem to be at the center of a large pool of elevated values from station 5 to 12, and observed down to depths of about 150 m. This area seems to match the salinity signature of the subtropical water pool in that part of the eastern Indian Ocean (see Fig. 2b in Parida and Antoine (2025)).

Missing data at stations 1-5 and station 8 prevents from getting a full transect for the backscattering coefficient, $b_{bp}(470)$ (Fig. 2d). The range of values is from 0.0004 to 0.0016 m$^{-1}$. They generally followed the chlorophyll-a distribution pattern, except at station 18, where no increase in $b_{bp}$ matches that of TChl-a. The larger backscattering values in this data set come from the deep chlorophyll maxima that developed around the euphotic depth.

The dominance of either PSC or PPC is illustrated on Fig. 2e, clearly showing that the latter are typical of surface waters (top ~50 m) of the most oligotrophic part of the transect (stations 7 to 17), where high irradiance and clear waters combine to trigger significant photoprotection.

The distributions of the 0-150 m data for TChl-a, POC, $c_p(660)$ and $b_{bp}(470)$ are displayed in Fig. 3 and the range of their values given in Table 1. The TChl-a values range from about 0.01 to 0.6 mg m$^{-3}$ (Fig. 3a), with a mode at ~0.2 mg m$^{-3}$, and with all values lower than about 0.05 mg m$^{-3}$ being for depths greater than 100 m. Fifty two percent of the values are for depths < 50 m. An about 10-fold variation in POC is observed from 6 to 70 mg m$^{-3}$ (Fig. 3b), with the mode for surface values (depths < 50 m) being about 30 mg m$^{-3}$. The mode of the $c_p(660)$ distribution (Fig. 3c) is around 0.06 m$^{-1}$, with very few data above 0.1 m$^{-1}$. The $b_{bp}(470)$ distribution (Fig. 3d) confirms the narrow range of values between about 4 10$^{-4}$ and 1.5 10$^{-3}$ m$^{-1}$ with a peak value at about 1 10$^{-3}$ m$^{-1}$.





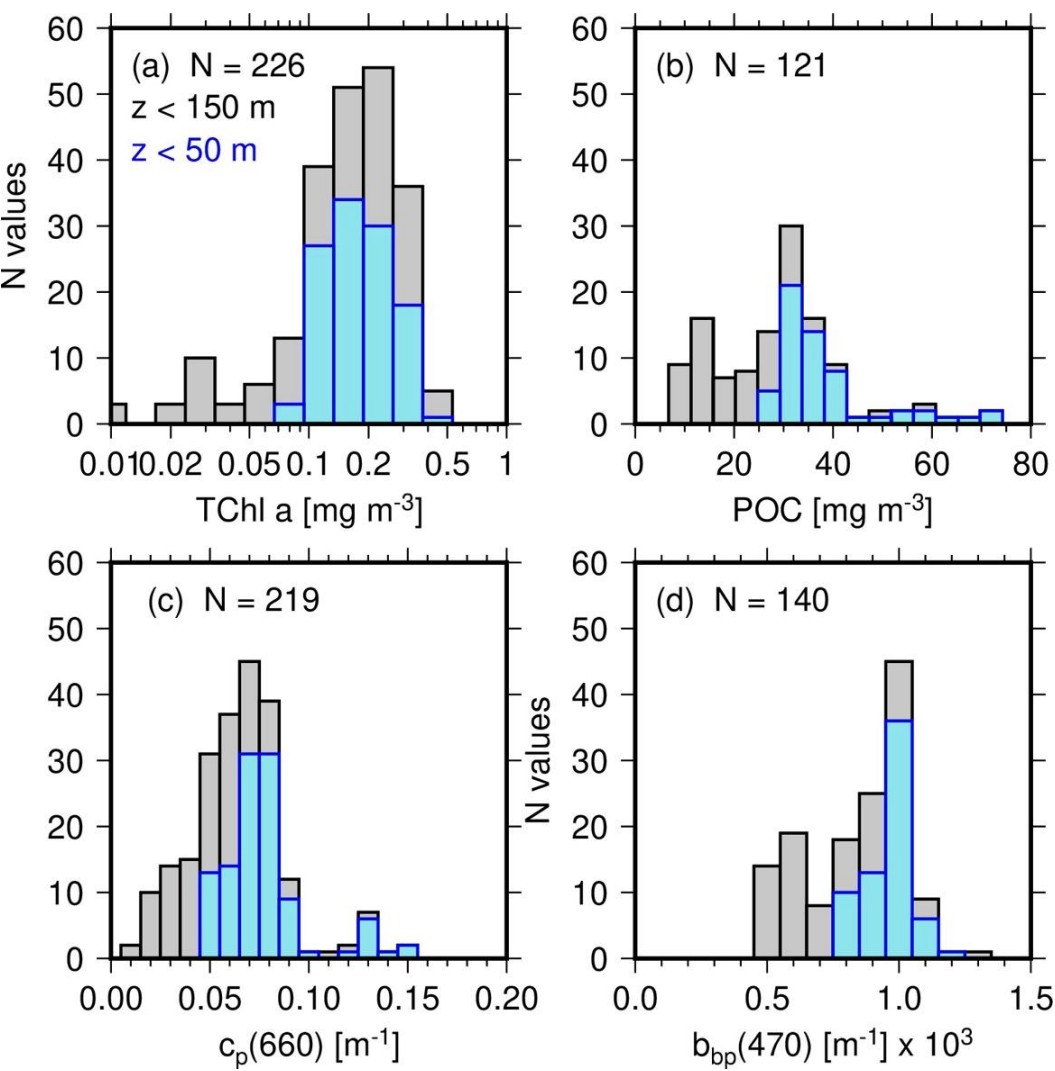

**Fig. 3**. Distribution of **(a)** total chlorophyll a concentration (TChl-a), **(b)** particulate organic carbon concentration (POC), **(c)** beam attenuation coefficient at 660 nm, $c_p(660)$ and **(d)** particulate backscattering coefficient, $b_{bp}$ (470), for the 20 stations of the 110°E voyage. The greyed histograms are for all data in the top 150 m and the blue for surface data only (depth < 50 m).





**Table 1**

Range of measured or calculated values for the parameters indicated.

| Parameter | Range of measured values | |
| --- | --- | --- |
| | Depths down to 150 m | Depths < 50 m |
| **POC (mg C m$^{-3}$)** | 6.8 – 70 | 25.7 – 70 |
| **TChl-a (mg Chl m$^{-3}$)** | 0.004 – 0.5 | 0.092 – 0.414 |
| **$b_{bp}$(470) (m$^{-1}$)** | 0.000456 – 0.00126 | 0.0008 – 0.00116 |
| **$c_p$(660) (m$^{-1}$)** | 0.014 – 0.15 | 0.047 – 0.15 |
| | **Range of calculated values** | |
| | Depths down to 150 m | Depths < 50 m |
| **$C_{phyto}$, from Chl vs. POC (mg C m$^{-3}$)** | 1.4 – 28.2 | 9.8 – 25 |
| **$C_{phyto}$ from $b_{bp}$(470) (mg C m$^{-3}$)** | 6.1 – 15.9 | 10.4 – 14.6 |
| **$C_{phyto}$ from cytometry (mg C m$^{-3}$)** | 0.145 – 100 | 0.95 – 100 |

### 3.2. POC and TChl-a vs. optical properties

Fig. 4 shows either POC or TChl-a as a function of $b_{bp}$(470) or $c_p$(660). These relationships are often represented in a log-log space when several orders of magnitude are covered by the data, with distributions close to log-normal. Because our dataset mostly includes oligotrophic waters and accordingly covers a small range of these properties, we choose to use a linear scale to display the data and to quantify their relationship through type II linear regression (least-square fit). The resulting slope and intercept of the fits are displayed in each panel of Fig. 4, along with the regression coefficient and RMSE of the fit (I units of POC or TChl-a). Ordinary least square Type II regressions were used ("lmodel2" R (1.7-2) function; Legendre (2014)). When the relationship between $C_{phyto}$ and either POC or TChl-a or IOPs was clearly not linear, however, we assessed their relationships in a log-log space. Regression statistics are all reported in Table 2.





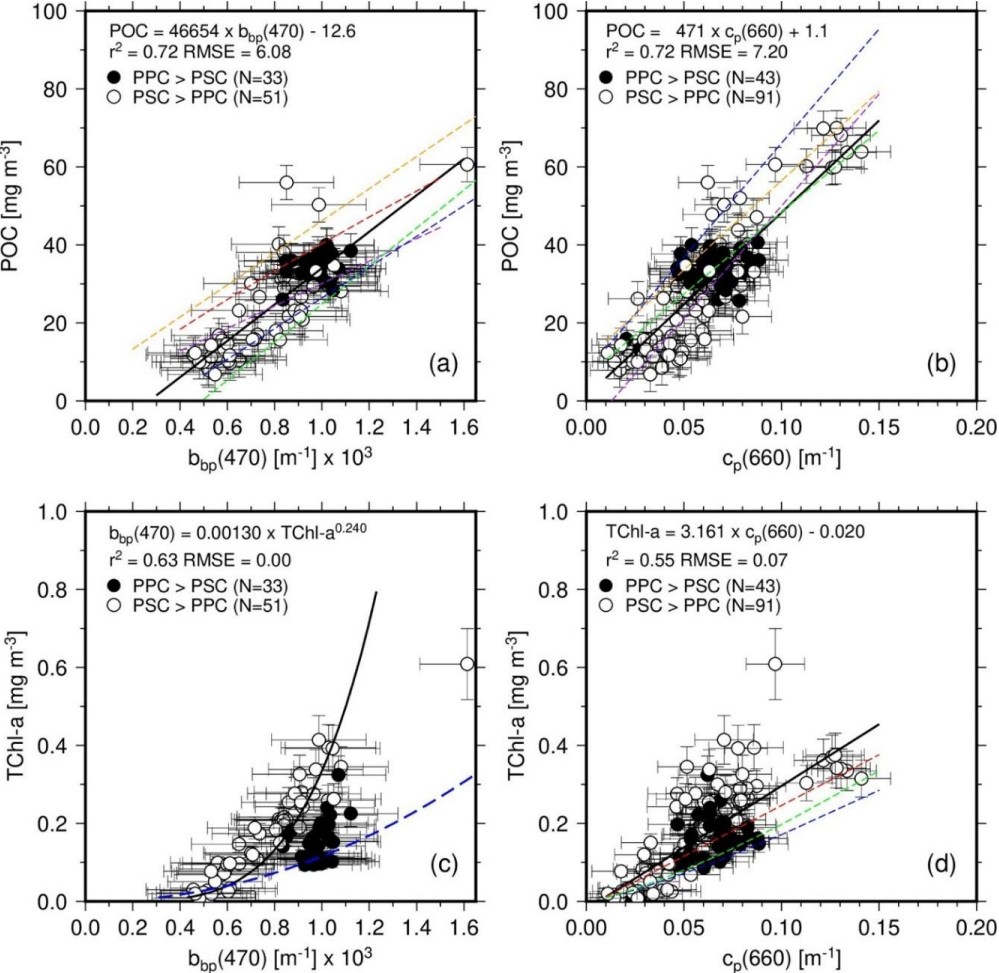

**Fig. 4. (a)** POC as a function of $b_{bp}(470)$ for depths < 150 m, and associated least square fit (thick black line; equation as indicated). Open symbols are used when PSC dominate PPC and black symbols otherwise. When the latter are ignored, the slope and intercept of the fit are only slightly changed (46,258 and -12.8). Previously established relationships are displayed as dashed lines: Stramski et. al., 1999 (red; POC = 17,069 $b_{bp}(510)^{0.859}$), Stramski et. al., 2008 (orange; POC = 53,932 x $b_{bp}(555)$ + 5.049), Loisel et.al., 2001 (purple; POC = 37,750 x $b_{bp}(555)$ + 1.3), Graff et. al., 2015 (green; POC = 48,811 x $b_{bp}(470)$ - 24), Thomalla et. al., 2017 (blue; POC = 39418 x $b_{bp}(470)$ - 13). **(b), (c) and (d)** As in (a) but for POC vs. $c_p(660)$, TChl-a vs. $b_{bp}(470)$ and TChl-a vs. $c_p(660)$. In (c) the fit is performed on the log-transformed data and with $b_{bp}$ being the quantity derived from TChl-a, for comparison with existing relationships (Antoine et al. (2011); dashed blue line). In (d), the dashed red line is from Behrenfeld and Boss (2006), the dashed blue from Loisel and Morel (1998) and the dashed green from Morel and Maritorena (2001) (for these two, their $c_p$ vs. Chl relationship was reversed).





385  **Table 2.** Outputs of the regression analyses. Results for both all data and PSC-dominated data only are provided when they are significantly different. None of these relationships are supposed to be applied to data sets collected in different environments.

| Parameter | panel | Regression analysis with $C_{phyto}$ derived from $b_{bp}$ | | | | |
|---|---|---|---|---|---|---|
| | | Slope | Intercept | $r^2$ | RMSE[a] | MAPE (%)[b] |
| **POC (mg C m$^{-3}$)** | 6c | 0.180±0.028 | 5.97±0.78 | 0.69 | 1.30 | 10.12 |
| **TChl-a (mg Chl m$^{-3}$)** | 6a | 14.58±3.13 | 8.40±0.61 | 0.38 | 1.80 | 15.43 |
| **$c_p$(660) (m$^{-1}$)** | 7e | 81.41±14.82 | 6.07±0.91 | 0.47 | 1.64 | 13.57 |
| | | **Regression analysis with $C_{phyto}$ derived from POC vs. Chl** | | | | |
| | | Slope | Intercept | $r^2$ | RMSE | MAPE (%) |
| **$b_{bp}$(470) (m$^{-1}$)**, *all data* | 7a | 18,720.8±3,462.2 | -2.10±3.0 | 0.45 | 3.88 | 29.18 |
| *PSC-dominated only* | | 29,552±3,154 | -8.60±2.49 | 0.81 | 2.81 | 21.63 |
| **$c_p$(660) (m$^{-1}$)**, *all data* | 7c | 151.68±20.12 | 4.45±1.39 | 0.51 | 3.73 | 30.60 |
| *PSC-dominated only* | | 167.3±21.50 | 4.40±1.48 | 0.63 | 3.70 | 30.91 |
| | | **Regression analysis with $C_{phyto}$ derived from cytometry** | | | | |
| | | A | B | $r^2$ | RMSE | MAPE (%) |
| **POC (mg C m$^{-3}$)** | 6d✤ | 0.0017±0.003✤ | 2.43±0.33✤ | 0.65 | 10.53 | N/A |
| **TChl-a (mg Chl m$^{-3}$)** | 6b✤ | 58.56±1.38✤ | 1.36±0.15✤ | 0.58 | 13.63 | N/A |
| **$b_{bp}$(470) (m$^{-1}$)** | 7b✤ | 7.39±1.21✤ | 4.15±0.61✤ | 0.57 | 8.05 | N/A |
| **$c_p$(660) (m$^{-1}$)** | 7d✤ | 2565.7±2.5✤ | 2.28±0.32✤ | 0.48 | 12.57 | N/A |
| | | **Regression analysis with POC** | | | | |
| | | Slope (or A) | Intercept (or B) | $r^2$ | RMSE | MAPE (%) |
| **TChl-a (mg Chl m$^{-3}$)** | 5a✤ | 0.570±0.11✤ | 80.64±1.23✤ | 0.45 | 9.96 | 28.37 |
| **$b_{bp}$(470) (m$^{-1}$)** | 4a | 46,654.5±6,447 | -12.58±5.60 | 0.72 | 6.08 | 19.55 |
| **$c_p$(660) (m$^{-1}$)** | 4b | 471.6±50 | 1.14±3.31 | 0.72 | 7.20 | 24.10 |
| | | **Regression analysis with TChl-a** | | | | |
| | | Slope (or A) | Intercept (or B) | $r^2$ | RMSE | MAPE (%) |
| **$b_{bp}$(470) (m$^{-1}$)**✩ | 4c | 1.3 10$^{-3}$±1.1 10$^{-4}$✤ | 0.241±0.04✤ | 0.63 | 0.0074 | N/A |
| **$c_p$(660) (m$^{-1}$)** | 4d | 3.161±0.48 | -0.018±0.03 | 0.55 | 0.07 | N/A |

✩  Here the regression is $b_{bp}$ *vs* TChl-a

390  ✤ In this case the coefficients reported are those of the form A x [independent quantity]$^B$

[a] Units of RMSE are those of the regressed parameter (so, either $C_{phyto}$, POC or TChl-a)

[b] MAPE is not reported for regressions performed on log-transformed data.




The error bars correspond to the uncertainties as previously described. It is noted that Fig. 4 and all subsequent Figures use data for the top 150 m, unless otherwise stated. The very small POC and TChl-a values from greater depths were indeed considered unreliable because too close from the detection limits and uncertainties of the measurement techniques. In Figs. 4 to 7, Open symbols are used when photosynthetic carotenoids (PSC) dominate over photoprotective pigments (PPC) and black symbols otherwise.

The relationship between POC and $b_{bp}(470)$ in our data set is shown on Fig. 4a. The PPC-dominated situations all correspond to surface waters (depth < ~50 m) from stations 7 to 17 and stations 19 and 20. They correspond to the most oligotrophic conditions encountered along the transect, with Chl-a < 0.1 mg m$^{-3}$. Data for deeper depths at these stations and for all depths at all other stations shows a dominance of PSC. The PPC-dominated points form a cluster centered around values of about 1 10$^{-3}$ m$^{-1}$ for $b_{bp}(470)$ and 40 mg m$^{-3}$ for POC. They do not stand out from the overall data cloud.

The linear regression in this dataset is displayed along with those derived by previous studies using data from other regions of the World oceans, using mostly surface data. To the best of our knowledge, no such relationship was established for the eastern Indian Ocean. The Loisel et al. (2001) (gold) relationship was derived from ocean color satellite observations of the Mediterranean Sea, the Stramski et al. (1999) (red) from field measurements taken in the Arctic Polar Front Zone, the Stramski et al. (2008) (orange) from the eastern south Pacific and Atlantic, the Graff et al. (2015) (green) from the Equatorial Pacific and the Atlantic Meridional transect, and the Thomalla et al. (2017) (blue) from the South-East Atlantic Ocean. These studies all used surface data sets (depth < ~40 m), and some of them use $b_{bp}$ at another wavelength than 470 nm. Therefore, we transferred our $b_{bp}(470)$ to these other wavelengths using Eq.3 with an exponent of -1.63, as determined from our data set. The appropriate equation was then applied and the resulting POC value plotted as a function of $b_{bp}(470)$. The Graff and Thomalla relationships are the closest to what we obtain here, although they rather follow the lower envelope of our data set. All relationships and the one derived here have similar slopes, with differences mostly within their uncertainties.

A significant correlation is also obtained for the POC vs $c_p(660)$ relationship (Fig. 4b), with a slightly larger RMSE than for POC vs $b_{bp}(470)$ relationship. Similarly to the $b_{bp}$ relationship, the cluster of PPC-dominated data does not depart from the general data distribution. The average $c_p(660)$ value for these data is 0.07 m$^{-1}$.

The PPC-dominated data however stands out when plotting TChl-a as a function of $b_{bp}(470)$ (Fig. 4c), indicating a significant non-algal contribution to backscattering. Note that in this panel, the regression uses TChl-a (vertical axis) as the predictor of $b_{bp}(470)$, to allow comparison with previously established similar relationships (see Figure legend). Finally, Fig. 4d shows the TChl-a vs. $c_p(660)$ relationship, with the smallest regression coefficient among the four relationships displayed in Fig. 4.

It is worth noting that, if only surface waters (depths < 55 m) are considered, the correlations of POC





and TChl-a with $b_{bp}(470)$ completely vanish, which is essentially due to the range of values becoming too small to allow any regression with POC or TChl-a (missing surface data for the mesotrophic waters). The correlations still hold for $c_p(660)$, with similar slopes yet lower correlation coefficients (0.62 for POC and 0.45 for TChl-a).

### 3.3. The three $C_{phyto}$ estimates

Contrary to what was said for Fig. 4, we here displayed POC as a function of TChl-a in a log-log space (Fig. 5a), to be consistent with many previous studies (Morel, 1988; Sathyendranath et al., 2009). The fit applied to their log-transform is displayed as the black solid line in Fig.5a. The slope and intercept are very close to those in the relationship reported by Morel (1988) (black dotted line), which was established for case-I waters with chlorophyll concentrations from 0.03 to 30 mg m$^{-3}$. The Sathyendranath et al. (2009) relationship (black dashed line) has a larger coefficient (180 instead of 80 here) and accordingly predicts higher POC values. This difference probably stems from their data set including meso- and eutrophic waters with much higher chlorophyll concentrations than what we have here.

A 1% quantile regression was conducted on these data to derive $C_{phyto}$ following Sathyendranath et al. (2009). The result is displayed as the green solid line on Fig. 4a. Similarly to the POC vs TChl-a relationship, it predicts lower $C_{phyto}$ than Sathyendranath et al. (2009) (dashed green line). The latter was however derived for data mostly from depths < 40 m. When the same depth limit is applied to our data set, the relationships come much closer one to each other (blue solid line and dashed green line in Fig. 5a), with comparable slopes and intercepts (see legend of Fig. 5).

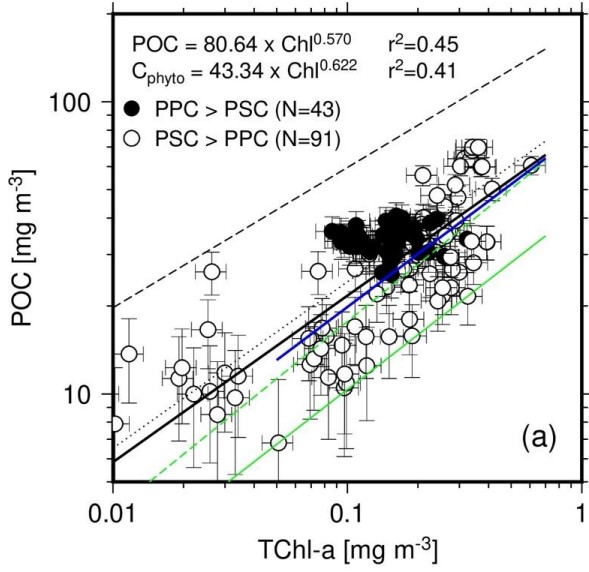

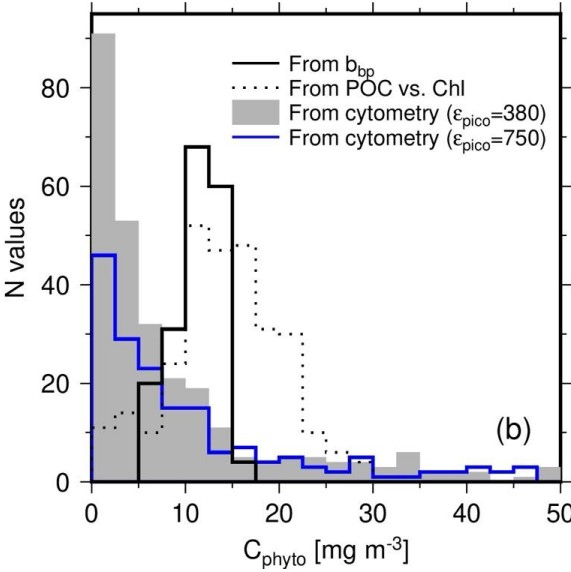



**Fig. 5. (a)** POC as a function of TChl-a for depths < 150 m and associated least squares fit to the log-transformed data (back thick line). Previously established POC vs. TChl-a relationships are displayed: Morel (1988) (black dotted line; POC = 90 x TChl-a$^{0.57}$) and Sathyendranath et al. (2009) (black dashed line; POC = 180 x TChl-a$^{0.48}$). The green solid line is the 1% quantile regression on our data set (equation reported on the graph), while the green dashed line is the 1% quantile regression from Sathyendranath et al. (2009) ($C_{phyto}$ = 79 x TChl-a$^{0.65}$). The blue solid line is the 1% quantile regression when data from only the upper 40 m are considered ($C_{phyto}$ = 79.3 x TChl-a$^{0.60}$). **(b)** Histograms of $C_{phyto}$ derived from the POC vs TChl-a relationship displayed in panel (a), and from the two other methods, as indicated. The blue line is for cytometry-derived $C_{phyto}$ using a large conversion factor for picoeukaryotes (750 instead of 380 fg C $\mu$m$^{-3}$).

The PPC-dominated data (black dots) show no correlation between POC and TChl-a, and slightly larger POC values on average (about 33.8 vs. 28.6) when the TChl-a range covered by the PPC-dominated data is considered (0.08 to 0.3 mg m$^{-3}$).

The distribution of the $C_{phyto}$ values derived through the 1% quantile regression is displayed in Fig. 5b (dotted line), along with the distribution of the same quantity derived either from $b_{bp}$(470) (using Graff et al. (2015); solid line) or from the cytometry data (greyed area). The Graff equation was chosen for the $b_{bp}$ relationship because it is, to the best of our knowledge, the only one derived from combining contemporaneous in-situ particulate backscattering measurements and direct phytoplankton carbon determination from field samples. The trophic conditions in the Graff data set are also quite similar to ours, with Chl-a < 0.3 mg m$^{-3}$. The cytometry-derived $C_{phyto}$ include many more low values (below about 5 mg m$^{-3}$) than the other two and also more higher values (> ~25 mgC m$^{-3}$). It is worth remembering that the histogram of the $b_{bp}$-derived $C_{phyto}$ (Fig. 5b) is just a linear translation of the $b_{bp}$ histogram. Similarly, the histogram of the Chl-derived $C_{phyto}$ is the translation of the Chl histogram through the power law displayed in Fig. 5a. Considering the uncertainty in such relationships (Fig. 4), the distribution of the resulting $C_{phyto}$ values is probably more relevant than the specific mode or median value of the distributions.

### 3.4. $C_{phyto}$ vs. TChl-a and POC

The $C_{phyto}$ values obtained from both $b_{bp}$(470) and cytometry data are displayed as a function of either Chl (Figs. 6a, b; linear scale) or POC (Figs. 6c, d; log scale). Regression lines are shown for either the entire data set (solid lines) or only the PSC-dominated data (dashed lines). We used a log space for the relationships involving the cytometry-derived $C_{phyto}$ because of both the larger range of values and the obvious non-linear behavior of the data when plotted as a function of either TChl-a or POC (see Supporting Information Fig. S2 for the same data plotted using a linear scale).







**Fig. 6.** **(a)** and **(b)** $C_{phyto}$ calculated from $b_{bp}(470)$ using Graff et al. (2015) or from cytometry data as a function of TChl-a. **(c)** and **(d)**: as in (a) and (b) but as a function of POC. The equation reported in panels (a) and (c) are for the PSC-dominated data only (dashed line). The blue lines are from Graff et al. (2015)(their Fig. 4).

The PPC-dominated data stand out of the general $b_{bp}(470)$-derived $C_{phyto}$ vs. Chl relationship (Fig. 6a) but do not when the same $C_{phyto}$ data are plotted as a function of POC (Fig. 6c). It is reminded here that the PPC-dominated data are for the upper 50 m of the most oligotrophic stations. This decorrelation likely results from



photo-acclimation jeopardizing the carbon vs. chlorophyll connection. On the contrary, the data for the PSC-dominated waters (open symbols) show a significant correlation ($r^2$ = 0.78) with TChl-a. This seems to

indicate that changes in TChl-a and $C_{phyto}$ are well related in those situations, which may as well be simply showing the correlation between $b_{bp}$ and TChl-a.

The relationships that Graff et al. (2015) derived from their analytical $C_{phyto}$ measurements are superimposed as blue lines. They show that our data set has a lower backscattering per unit chlorophyll (Fig. 6a). If this ratio would be similar in both data sets, the blue line would be close to the regression line we

obtain here. A similar observation can be made for the $b_{bp}$-to-POC ratio (Fig. 6c), although the slopes of the relationships are much closer than for Chl.

The cytometry-derived $C_{phyto}$ also generally increase with TChl-a and POC (Fig. 6b, d). The exponent of the power law is nearly twice as large for the relationship with POC as compared to the relationship with TChl-a.

**3.5. $C_{phyto}$ vs. optical properties**

Particle backscattering is largely influenced by submicrometer particles (Ulloa et al., 1994), when beam attenuation is rather sensitive to the size range ~0.5 to 20 μm (Boss et al., 2001). They are both sensitive to the particle size distribution. Therefore, examining the relationship between the three $C_{phyto}$ estimates with these two optical properties might give us some clues about the influence of the particle assemblage. This is

illustrated on Fig. 7.



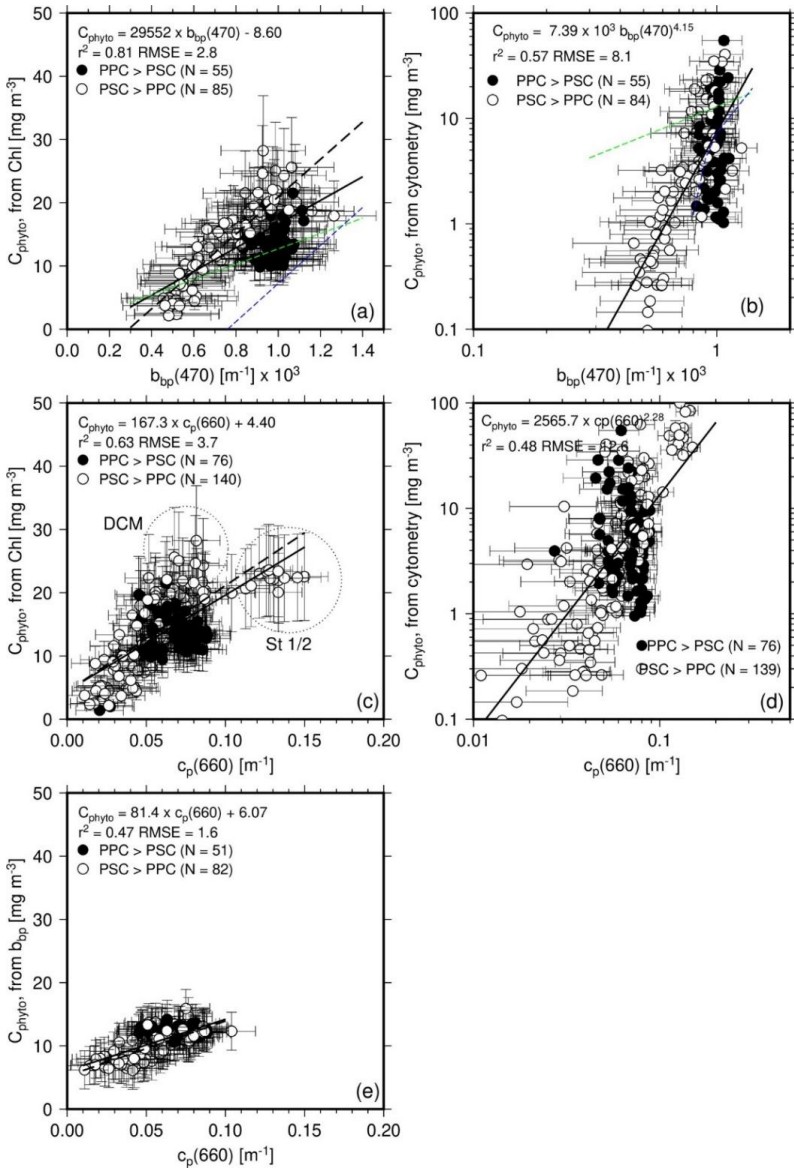

**Fig. 7. (a)** $C_{phyto}$ calculated from the POC vs. Chl-a relationship as a function of $b_{bp}(470)$. The solid black line is the regression line using all data points, while the black dashed line is for the PSC-dominated data only. The Graff et al. (2015) relationship is displayed as the dashed green line and the Martinez-Vicente et al. (2013) as the dashed blue line. **(b)** $C_{phyto}$ calculated from cytometry data as a function of $b_{bp}(470)$. No regression was attempted with these data that seem poorly correlated. Only the Graff and Martinez-Vicente relationships are superimposed for reference. **(c)** and **(d)**: as in **(a)** and **(b)** but as a function of $c_p(660)$. **(e)** $C_{phyto}$ calculated from $b_{bp}(470)$ as a function of $c_p(660)$. No points appear for $c_p > \sim 0.1$ because no $b_{bp}$ data were simultaneously measured (stations 1-4).






The $C_{phyto}$ values derived from the 1% quantile POC vs. Chl regression are displayed as a function of the measured $b_{bp}(470)$ in Fig.7a along with the least-square fit to all data (black continuous line) and to PSC-dominated data only (black dashed line). The Martinez-Vicente et al. (2013) relationship is displayed as a dashed blue line, and that of Graff et al. (2015) as a dashed green line. The cluster of PPC-dominated data (black symbols) does not show a clear relationship with $b_{bp}$, and separates from the PSC-dominated data, with lower $C_{phyto}$ values for given $b_{bp}$ values. The slope of the Graff equation is lower than what we derive here, while the relationship of Martinez-Vicente et al. (2013) has a closer slope to ours yet a large negative intercept, leading to much lower $C_{phyto}$ predictions. The Martinez-Vicente et al. (2013) relationship was based on measurements from the first optical depth along the Atlantic Meridional Transect cruise, using $C_{phyto}$

estimates derived from flow cytometry. The slope of the Graff et al. (2015) relationship is closer to what we get for the PPC-dominated situations, although the correlation is weaker in our case.

Fig. 7b displays the same comparison as Fig. 7a but for the $C_{phyto}$ derived from cytometry data and uses a log scale. The cytometry-derived values span a larger range than those derived from the POC-based approach, with many more low values and some higher values as well. The power law derived from

regression on the log-transformed data shows a large exponent. It can be noted, however, that the Martinez-Vicente et al. (2013) equation (blue dashed line) is somewhat more consistent with what we derived here from cytometry data as well.

Results are rather similar when the two $C_{phyto}$ estimates are plotted as a function of $c_p(660)$ (Figs. 7c, d). The largest Chl-derived $C_{phyto}$ (between ~25 – 30 mg m$^{-3}$; Fig. 7c) do not correspond to the largest $c_p$ values,

which are observed at stations 1-2 ($c_p > $~0.1 m$^{-1}$ and $C_{phyto}$ ~22 mg m$^{-3}$) where Chl is about 0.3 mg m$^{-3}$, and rather to data from the deep-chlorophyll maximum at stations north of station 12 (and except station 18). This is because the $c_p$/Chl ratio is larger at stations 1-2 than it is for the DCM at the other stations (0.39 vs. 0.18 m$^2$ mg$^{-1}$), as a result of photo-acclimation. The cytometry-derived $C_{phyto}$ values do show a correlation with $c_p(660)$, again when established in the log space (Fig. 7d).

The $b_{bp}$-derived $C_{phyto}$ show a rather tight relationship with $c_p(660)$ (Fig. 7e), and no specific behavior of the PSC- or PPC-dominated data, owing to a correlation of backscattering and attenuation for the entire data set.

## 4. Discussion

The main purpose of this study was to assess three methods to derive $C_{phyto}$ in oligotrophic waters, when

direct measurements are not available. One of these methods uses $b_{bp}$, suggested as a relevant optical proxy to $C_{phyto}$ by Behrenfeld et al. (2005). The second method uses the POC vs. Chl relationship, following Sathyendranath et al. (2009). Some studies have indeed shown chlorophyll as a relevant proxy for



phytoplankton carbon or photosynthetic activity (e.g., Huot et al. (2008)), yet photo-acclimation may lead to erroneously interpret temporal changes in Chl at a given site or changes along the vertical in the water column

as changes in $C_{phyto}$. The third method derives $C_{phyto}$ from cell counts and allometric conversion factors. Although this method is theoretically less subject to overestimation by incorrectly accounting for non-algal material, it may on the contrary underestimate $C_{phyto}$ by missing either very small or large cells or both. It also relies on conversion factors that are poorly constrained.

From the comparison of these three methods, we cannot unambiguously conclude as to which one

generates more accurate $C_{phyto}$ values because we did not have direct measurements to compare with. Nevertheless, the comparison of the $C_{phyto}$ values they produce and of their relationships with optical and biogeochemical quantities can provide some clues on the relevance of these techniques to assess $C_{phyto}$ distributions in oligotrophic waters.

Here, we first discuss the data set we worked from, then we compare the three $C_{phyto}$ estimates, assess their

dependence on the phytoplankton pigments and community, and finally discuss the implication of our results for the derivation of $C_{phyto}$ from satellite ocean color observations.

### 4.1. An oligotrophic data set complementary to previous studies

The relationships we observe between biogeochemical quantities (POC and TChl-a) and inherent optical properties ($b_{bp}$(470) and $c_p$(660)) are generally consistent with similar relationships derived by previous

studies (Fig. 4). This shows that the bio-optical characteristics in the oligotrophic waters of the eastern Indian Ocean do not obviously depart from what is observed in other oceans. A higher backscattering per TChl-a is however observed for surface waters (depth < ~50 m) of the central part of the transect (black dots in Fig. 4c). A larger contribution on non-algal particles (NAP) could explain this, although not confirmed by a larger NAP absorption. The cluster of points with the higher backscattering per TChl-a indeed corresponds to the

absorption-based cluster G2 in Parida and Antoine (2025), which does not show high NAP absorption (their Fig. 8). This cluster is however the one with the highest PPC concentration. This higher PPC concentration associated to a larger backscattering could reveal the presence in the phytoplankton population of smaller, more refringent cells.

The exponent of the POC vs. TChl-a relationship (0.572; Fig. 5a) is identical to the one derived by Morel

(1988) and slightly higher than what Sathyendranath et al. (2009) obtained (0.48). The exponents of the 1% quantile regression are very close (0.62 here vs. 0.65). The coefficient of this relationship in Sathyendranath et al. (2009) is nearly twice the one we obtain here (79 vs. 43), however, because their data set includes much larger POC and TChl-a concentrations. The consistency with the Morel relationship, established for open ocean Case I waters, is another indication that the waters we have sampled belong to this category and have

rather average bio-optical characteristics. The large coefficient difference with Sathyendranath et al. (2009)



emphasizes that a ubiquitous POC vs TChl-a does not seem to hold, as often reported (e.g., Lee et al. (2020).

### 4.2. Comparison of the three $C_{phyto}$ estimates

The $C_{phyto}$ values derived from our POC vs. TChl-a relationship show a near normal distribution (Fig. 5b) with a mode value around 15 mg m$^{-3}$ (see range in Table 1). The distribution is quite similar for the $b_{bp}$(470)-
derived $C_{phyto}$ values, although the range is smaller and maximum values are lower (~16 mg m$^{-3}$ instead of ~30 mg m$^{-3}$). The values derived from cytometry data show a very different distribution skewed towards low values (55% < 5 mg m$^{-3}$). Figs 6, 7 and Sup Fig. 2 show that the cytometry-derived $C_{phyto}$ is not linearly related to POC and TChl-a neither it is to the IOPs ($b_{bp}$ and $c_p$), which are quantities determined by both phytoplankton and non-algal particles. This indicates that particles unaccounted for by cytometry do not
covary with the phytoplankton identified through this method. Non-algal particles being generally found covarying with phytoplankton, this would rather indicate that the particles missed by cytometry are rather small and large phytoplankton. The slopes of the linear regressions between the cytometry-derived $C_{phyto}$ and IOPs in the log space (Fig. 7) are around 4.5 for $b_{bp}$ and 2.5 for $c_p$, with correlation coefficients around 0.5-0.7. This seems plausible considering that the cytometry-derived $C_{phyto}$ is essentially a linear function of the
total volume of particles (cubic function of size) when the IOPs are a function of the particle size, among other parameters such as the particle composition. This compound of observations confirm that the cytometry-derived $C_{phyto}$ is, as intended, phytoplankton-specific (i.e., not including non-algal material). The Chl- and $b_{bp}$-derived $C_{phyto}$ are linearly related to the IOPs, showing that both estimates are affected by non-algal material.
This comparison is further illustrated on Fig. 8, showing weak correlation between $b_{bp}$-derived and Chl-derived $C_{phyto}$ values, with a $r^2$ of 0.55 and a slope of their relationship of 1.61. There is no correlation at all when the cytometry-derived $C_{phyto}$ data are considered. The dark symbols in Fig. 8 for Chl-derived $C_{phyto}$ look however like a subset of the data cloud formed by the cytometry-drived $C_{phyto}$ (open symbols). This may indicate that the lowest $C_{phyto}$ values, assuming they are realistic, cannot be derived when using Chl because
the method still accounts for non-algal material. The same result may however rather show that the cytometry method underestimates $C_{phyto}$ in the clearest waters by missing part of the phytoplankton population. The limited capability of the $b_{bp}$-based method to reproduce the variability in $C_{phyto}$ that the two other methods produce, in particular the cytometry-based method, is also clearly illustrated by Fig. 8.




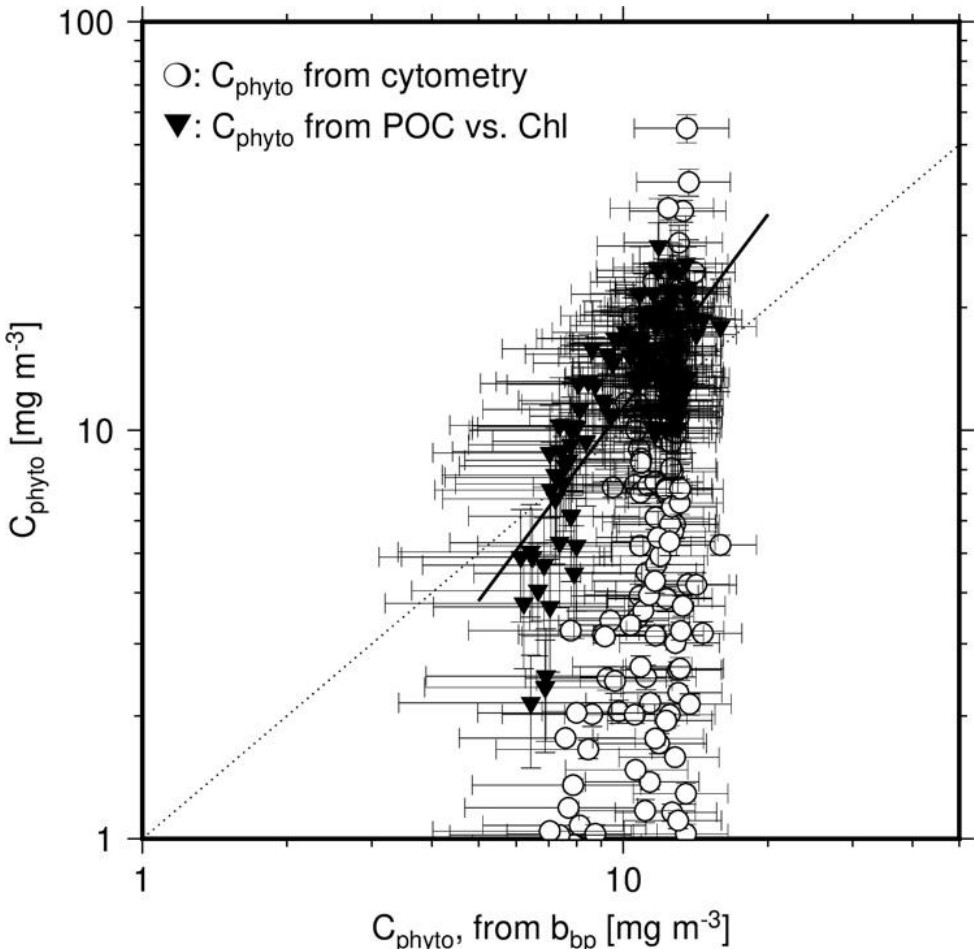

**Fig. 8.** Scatter plot of $C_{phyto}$ derived either from cytometry data (open circles) or from the POC vs. Chl relationship (1% quantile regression derived from our data set; black inverted triangles), as a function of $C_{phyto}$ derived from $b_{bp}(470)$, using Graff et al. (2015). The black line is the linear regression on the log-transformed data for $C_{phyto}$ derived from the POC vs. Chl relationship vs. $C_{phyto}$ derived from $b_{bp}(470)$. The dotted line is the 1:1 line.

### 4.3. Community composition impact on $C_{phyto}$ estimation

The influence of community composition on $C_{phyto}$ estimation was essentially examined through the relative contributions of PSCs and PPCs, which play critical roles in determining phytoplankton function and resilience through influencing both the physiological state and optical properties of phytoplankton.

The results showed that PPC-dominated waters, essentially surface low-chlorophyll waters, have higher $b_{bp}$ per unit Chl-a. Conversely, higher PSC levels correspond to increased $C_{phyto}$, reflecting greater biomass production. Environments with low Chl-a concentrations and higher proportion of PPC typically represent



adaptations to high light or nutrient-limited environments, where photoprotection is crucial (Fig 2e). Under these conditions, $C_{phyto}$ per cell tends to be lower, as the physiological makeup prioritizes the production of PPC over cellular carbon biomass, resulting in lower carbon-to-chlorophyll ratios. Conversely, regions with

higher PSC levels indicate higher photosynthetic activity (higher POC), often corresponding to nutrient-rich conditions or the need to increase the PSC pool in low-light conditions. Field studies indicate that phytoplankton consistently make up a steady portion of POC across seasons and varying conditions from oligotrophic to eutrophic conditions, with some exceptions (Behrenfeld et al., 2005). In oligotrophic regions, phytoplankton typical constitute a smaller fraction of POC pool compared to more productive, nutrient-rich

environments (Negrete-García et al., 2022). This is largely due to the greater contribution of detrital material and dissolved organic matter in oligotrophic environments. Furthermore, shifts in pigment composition (ratio of PSC and PPC) can influence cellular carbon content, thereby affecting the overall phytoplankton associated POC. In such a scenario, estimating $C_{phyto}$ using the 1% quantile regression of POC: Chl-a data presents limitations as POC can vary independently of chlorophyll-a concentrations.

The contribution of PSC and PPC also significantly affect $C_{phyto}$ derived from $b_{bp}$. Higher $C_{phyto}$ values were observed when the contribution of PPC was greater compared to PSC (Fig 6a). This pattern is especially evident at stations 12–17 in oligotrophic regions, where elevated surface $C_{phyto}$ computed from $b_{bp}$ corresponds with high PPC concentrations (Fig. 7a). A possible cause of elevated $b_{bp}$ value is the higher relative abundance of picoplankton, which is a typical feature of oligotrophic waters. These environments

are typically subject to chronic nutrient limitation, which promotes the dominance of small picophytoplankton such as Prochlorococcus and *Synechococcus* (Flombaum et al., 2013), These small cells generally contribute to increased $b_{bp}$ (Brewin et al., 2012) and, consequently, higher estimates of $C_{phyto}$. Although the $b_{bp}$ method has broad application and essential for estimating $C_{phyto}$ from the ocean color satellite data, its effectiveness is limited when other optically active substances significantly contribute to

$b_{bp}$, making it difficult to isolate the phytoplankton-specific backscattering signal.

We observed that phytoplankton community composition had minimal impact on $C_{phyto}$ estimates derived from flow cytometry. In some way, cytometry is a better option for compute $C_{phyto}$. However, the approach also involves several methodological uncertainties and limitations. Converting cell counts to carbon biomass typically relies on empirical allometric relationships, which may not adequately capture species-specific

variability or physiological responses to environmental conditions. The estimation of cell size from forward scatter (FSC) is influenced not only by cell volume but also by refractive index and internal structural complexity, leading to potential inaccuracies (Olson et al., 2018). As flow cytometry typically targets cells in the picophytoplankton and nanophytoplankton size classes. It may systematically underestimate $C_{Phyto}$ when large cells contribute significantly especially relevant in high-nutrient environments. These findings

highlight that bio-optical relationships should be region-specific to reduce biases and uncertainties. The



methodology and local environmental conditions, such as phytoplankton community composition, play a significant role in shaping these relationships. Therefore, empirical bio-optical models should be tailored to the specific regions in which they are applied to ensure accuracy.

### 4.4. Implications on deriving $C_{phyto}$ from satellite ocean color observations

The $b_{bp}$- and Chl-based methods used here to derive $C_{phyto}$ are applicable to satellite ocean color observations and have actually been proposed for that purpose. Therefore, we have compared their average outputs for the layer which the remote sensing signal originates from in clear waters. This depth is roughly the inverse of the diffuse attenuation coefficient for downward irradiance (Gordon and Mccluney, 1975). Because we used discrete data at depths of about 5, 10 and 20 m, we calculated equivalent satellite-derived

values as the average of the sum of three times the 5-m value plus two times the 10-m value plus the 20-m value. This is a way to approximate the optically-weighted Chl, accounting for the exponential decrease of light with depth that makes shallower values having more weight in forming the water-leaving radiance.

By construction, the $C_{phyto}$ derived from the POC vs. TChl-a relationship follows the TChl-a changes, (Fig. 9). The $b_{bp}$-derived $C_{phyto}$ remains quite steady, with values around 12-15 mg m$^{-3}$. This is directly related to

$b_{bp}(470)$ showing only an about 15% variation across the transect (except at station 18), when Chl varies by a factor of about 2.5 at surface over the part of the transect where both were measured. If the $b_{bp}$ vs TChl-a relationship (Fig. 4c) is used to reconstruct $b_{bp}$ values that could have been measured for stations 1 to 4, the associated range in $C_{phyto}$ still remains low. Therefore, if a true variability exists in $C_{phyto}$ along the transect, the particulate backscattering coefficient does not seem to be a relevant indicator of this variability. This is

an indication that phytoplankton play a too-small role in determining $b_{bp}$ for it to be used as a proxy to $C_{phyto}$. The NAP contribution to $b_{bp}$ could be as high as 60 to 80% in that area (e.g., Fig. 2 in Bellacicco et al. (2018)) and in oligotrophic areas in general (Bellacicco et al., 2019).





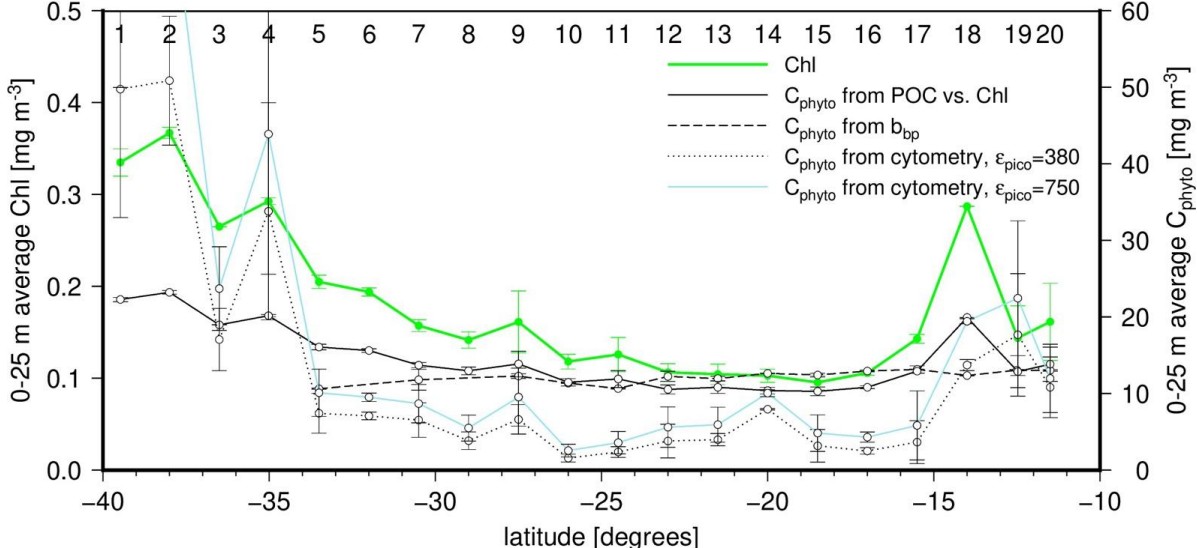

**Fig. 9.** Latitudinal distribution of the average concentration within the upper 25 m for chlorophyll (left Y-
axis; green line) and $C_{phyto}$ as derived from the three methods tested here (right Y-axis; line types as
indicated). The vertical bars are the standard deviations within surface data for the two casts at each station.
Station numbers are indicated on top of the panel

The cytometry-derived values display the strongest latitudinal changes, following the TChl-a changes.
This suggests that this technique indeed avoids accounting for non-algal material in determining $C_{phyto}$. It is
unclear from previous literature whether the low values ($< \sim$5 mg m$^{-3}$) we derived here are realistic,
however. Similarly low values were obtained by Martinez-Vicente et al. (2013), who also used cytometry
(their Fig. 2A). Their equation (2) gives a $C_{phyto}$ of 7 mg m$^{-3}$ for $b_{bp}(470) = 1 \times 10^{-3}$, with 70% of our
$b_{bp}(470)$ being lower than this value. Therefore, the question arises as to whether the cytometry technique
misses a significant part of the phytoplankton population or, on the contrary, the two other techniques
inevitably overestimate $C_{phyto}$ in oligotrophic waters because they rely on parameters that are either not
strongly-enough linked to phytoplankton carbon ($b_{bp}$) or have a variability that is somewhat disconnected
from changes in phytoplankton carbon because of photo-acclimation (TChl-a). It is reminded here that the
contribution to $C_{phyto}$ of large cells that escape the cytometry measurements is empirically derived through
the $f_{fc}$ factor (Eq. 6). Arbitrarily adjusting $f_{fc}$ to increase the cytometry-derived $C_{phyto}$ values for the most
oligotrophic waters so they match the other estimates would require unrealistically low $f_{fc}$ values (meaning
a very low contribution of pico-phytoplankton to $C_{phyto}$). Among the other parameters of Eq. (5), the cell
diameters are rather well constrained, and we have no indication that the cytometer counts could be largely
erroneous. Therefore, the conversion factors remain, as expected, the main source of uncertainty. The



dashed blue line in Fig. 9 shows how the cytometry-derived $C_{phyto}$ values are increased when a higher conversion factor for pico-phytoplankton is used (750 instead of 380 fg C $\mu m^{-3}$; see methods).

### *Data Availability Statement*

The 110E research voyage data (code IN2019_V03) are available on the CSIRO Marlin data base
(https://marlin.csiro.au) and on demand from the authors.

### *Conflicts of interest*

The authors do not have conflicts of interest to report.

### *Authors contributions:*

**David Antoine**: Conceptualization (equal); Formal analysis (equal); Funding Acquisition (lead); Investigation (lead); Methodology (equal); Project Administration (lead); Resources (lead); Software (equal); Supervision (lead); Validation (equal); Visualization (equal); Writing – Review and Editing (equal). **Chandanlal Parida**: Conceptualization (equal); Formal analysis (equal); Methodology (equal); Software
(equal); Validation (equal); Visualization (equal); Writing – Review and Editing (equal). **Camille Grimaldi**: Investigation (supporting); Writing – Review and Editing (supporting).

### Acknowledgments

This research was supported by the Australian Government through the Australian Research Council's
Discovery Projects funding scheme (project DP210101959). It is a part of Australia's contribution to the Second International Indian Ocean Expedition-(IIOE-2) and was supported by a grant of sea time on RV Investigator from the CSIRO Marine National Facility (https://ror.org/01mae9353). We also received funding from the European Space Agency for the collection of optical data. Peta Vine is thanked for her help with sample collection, filtration work and for the cytometry data collection and processing during the
voyage. Matthew Slivkoff and Wojciech Klonowski, In-situ Marine Optics Pty Ltd, Perth, are thanked for their help with collection of the optical data. The phytoplankton pigment analyses were carried out by Céline Dimier and Joséphine Ras from the "Service d'Analyses de Pigments par HPLC" (SAPIGH) analytical platform of the Institut de la Mer de Villefranche (CNRS-France).



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
