# Peer review of "Potential of optical and ecological proxies to quantify phytoplankton carbon in oligotrophic waters"

_EGUsphere, 2025_

## Author Comment (AC2)

Review of Antoine et al. 'Potential of optical and ecological proxies to quantify phytoplankton carbon in oligotrophic waters.

| Reviewer comment |
|---|
| This article explores the use of various approaches to estimate phytoplankton carbon across a range of waters in the Indian Ocean, with a focus on chlorophyll a, optical backscatter and flow cytometry as predictor variables. Such Cphyto estimates are of great value in marine biogeochemical studies, with important applications to satellite-based measurements. The results presented in this paper show a range of relationships between Cphyto and the various input variables, with significant correlations observed for the full data set, and weaker relationships for surface (<25 m depth) data.
 Overall, I think that this a valuable and interesting study, with important implications for the field.  One significant limitation, however, is the lack of a 'true' (i.e. gold-standard) measurement of Cphyto.  Without this validation, it is not possible to say which method provides the best approximation for Cphyto, as the authors themselves acknowledge. Nonetheless, I think the paper is still useful, as we can (with some modifications to the current text) get a sense of how the different proxies produce different Cphyto estimates.  Other things I note, is the need for a bit more discussion on the role of light-acclimation in driving some of the observed variability, more discussion of the size biases in the cytometer data, and a more robust application of the pigment data to discuss the role of phytoplankton taxonomy. |

| Authors response |
|---|
| We are pleased that this reviewer found our work valuable and important for the field and also acknowledged that we have clearly stated the limitations caused by the absence of direct analytical measurements of phytoplankton carbon.
 We have tried to improve our discussions to address the points raised here by the reviewer, whom we thank for a constructive and helpful review. |

Specific comments are listed below.

Specific comments:
**Reviewer comment or question**: Line 13: include physiological status as a source of variability
**Response:** done

**Reviewer comment or question**: Line 16: 'both are yet each in'??  check grammar here.
**Response:** the confusion comes from a misplaced comma. The sentence has been rewritten as follows and should be clearer now.

It is accordingly still unclear which of Chl-a or $b_{bp}$ is best suited to quantify $C_{phyto}$ or whether they both are, yet each in specific trophic conditions, especially for low-productivity oligotrophic waters.

**Reviewer comment or question**: Line 36: 'are' missing before 'accordingly'.
**Response:** corrected.

**Reviewer comment or question**: Line 63: reference after 'change significantly'
**Response:** Two references have been added:

Serra-Pompei, C., Hickman, A., Britten, G. L., & Dutkiewicz, S. (2023). Assessing the potential of backscattering as a proxy for phytoplankton carbon biomass. Global Biogeochemical Cycles, 37(6), e2022GB007556.

Xu, Wenlong, et al. "Spatiotemporal variability of surface phytoplankton carbon and carbon-to-chlorophyll a ratio in the South China Sea based on satellite data." Remote Sensing 13.1 (2020): 30.

**Reviewer comment or question**: Line 66.  I find the transition to this new paragraph a bit abrupt
**Response:** we have modified the starting sentence as follows:

Understanding these dynamics is particularly important in regions such as the Eastern Indian Ocean (EIO), which provides food, natural resources and numerous benefits to surrounding countries (Hermes et al., 2019). The EIO encompasses diverse hydrographic regimes that strongly influence phytoplankton productivity and physiology. In the northern EIO, the Bay of Bengal experiences monsoon-driven seasonal circulation changes during summer (southwest monsoon) and winter (northeast monsoon) (Schott and Mccreary Jr, 2001), along with large freshwater inputs from the rivers and rainfall that create surface stratification and barrier layer (Vinayachandran, 2009).

**Reviewer comment or question**: Last line of introduction – do you relate the results to environmental variables also?
**Response:** No, we have not related the results to any variables. However, in another publication (Parida et al., 2025), we have linked some of the results with environmental variables in the relevant sections.

**Reviewer comment or question**: Line 93: replace 'going' with 'sent'
**Response:** done.

**Reviewer comment or question**: Lines 94/95. Given the significant latitudinal gradient sampled, did these sampling times represent a consistent part of the diel cycle (e.g. xx hours after sunrise or sunset). If not, is it necessary to consider this?
**Response:** yes, we stayed on the same meridian, and these times always corresponded to dawn and dusk.

**Reviewer comment or question**: Line 106: was the filtration for pigments conducted under low light?
**Response:** yes, it was. This is now said.

**Reviewer comment or question**: Line 111: I think the 'P' in HPLC stands for 'performance'
**Response:** Yes, indeed. Corrected.

**Reviewer comment or question**: Line 169: I think it's important to provide more information on the size cutoffs of the instrument (lower and upper). What part of the size spectrum is being missed? This is mentioned very briefly on line 240, but it would be good to see it here, and with an upper cutoff also.
**Response:** we have added further information at the end of the paragraph here pinpointed.

**Reviewer comment or question**: Line 200: I would have thought that the value of gamma differed significantly between the different phytoplankton assemblages, based on their size spectra. What was the relative error in the mean gamma value, averaged for all samples?
**Response:** our data set covers a limited range of variability of phytoplankton assemblages. Therefore, the impact of their variability is not strong enough to generate substantial changes in the spectral slope of the backscattering coefficient (especially when the role of phytoplankton in $b_{bp}$ is rather low).

**Reviewer comment or question**: Line 208: What was the vertical resolution / sampling frequency of vertical bbp measurements?
**Response**: The Hydroscat sampling frequency is 1 Hz, and the profiling speed was about 0.25 m s$^{-1}$. We then had a vertical resolution of about 0.25 m (added in the text).

**Reviewer comment or question**: Line 213: I don't quite understand this method, based on POC chl regressions. It seems to me that the derived relationship would produce an average C:Chl ration, including a lot of detrital matter. It makes more sense after looking at figure 5, which could be cited here.
**Response:** here we refer the reviewer to the original paper by Sathyendranath et al (2009) and indeed, Fig. 5 probably makes quite clear what the logic is. We want to add here that we did not consider any of the tested methods, including this one, as necessarily exempt from uncertainty or underlaid by "rock solid" assumptions. It is precisely the goal of the paper to illustrate these uncertainties.

**Reviewer comment or question**: First paragraph p. 10. At this point, I was wondering about the various error terms. These are addressed below, but it might be good to at least mention this here.
**Response**: we have added ", and their uncertainties assessed later (section 2.6)." at the end of the first paragraph of section 2.5.

**Reviewer comment or question**: Line 245: What is the theoretical basis underlying the relationship between the slope of the size distribution and absorption at 676? Does it relate to pigment packaging? Some more information would be helpful for non-specialists.
**Response:** The rationale is exposed in Roy et al (2013) (their Eq. 10). The connection between the two exists because specific absorption of chlorophyll-a is a function of the cell diameter and the spectral dependence of backscattering has been shown a function of the particle size distribution.

**Reviewer comment or question**: I gather that there were no size-fractionated chlorophyll data? Those would have been really helpful to validate some of these results.
**Response:** indeed, we did not perform size-fractionation.

**Reviewer comment or question**: Line 304: It's true that conditions were more oligotrophic, but it's worth mentioning the significant sub-surface chl maximum, which had chl values higher than observed to the south.
**Response**: we are unsure what to answer here. We indeed mention this in the text. Or maybe we do not understand the comment.

**Reviewer comment or question**: Fig. 2. The red starts are not labelled in the legend (only in the main text). I would add dots to show the actual sampling points uses for the interpolations. For the bottom panel, rather than repeating chl, why not plot, for example, the PPC/PSP ratio, or POC:Tchl, which is mentioned in the text.

**Response:** The meaning of the red stars is now added. Note that the dots in panel (e) correspond to the sampling depths for each station so we do not repeat them in the other panels.

**Reviewer comment or question**: Last paragraph on p. 14. This is very descriptive material, which I think could be removed, as it's apparent from the figures.
**Response:** it is indeed somewhat descriptive, but this is the results section, so we decided to keep it as it is.

**Reviewer comment or question**: Fig. 3 could go in a supplement, I think.

**Response:** we prefer to keep it in the main text because from your comments and those of the other reviewer, it seems important to make clear that the data set covers a rather limited range of values, because mostly from oligotrophic waters.

**Reviewer comment or question**: Line 370: Cphyto is listed here as having a non-linear relationship, but that variable is not shown in the plots.

**Response**: yes, indeed, that is unclear. This sentence is not only for $C_{phyto}$. We have rewritten as follows:

When relationships among the various parameters here assessed were clearly not linear, we assessed them in a log-log space.

**Reviewer comment or question**: Fig. 4 bottom left panel: the white points look rather non-linearly distributed.
**Response**: yes, and that is why we used a power law to describe the relationship.

**Reviewer comment or question**: Table 2: I don't understand the last sentence in the table header.  Maybe change 'panel' to 'figure' along the column headers.
**Response:** what we try to say is that the relationships derived in this work are specific to our data set so great caution should be used if they are applied to predict, e.g., $C_{phyto}$ from IOPs measured in other environments. We have rewritten as:
"None of these relationships are supposed to be applied to data sets collected in environments markedly different from what we encountered during this IIOE voyage."
Otherwise, we have indeed changed "panel" to "Figure".

**Reviewer comment or question**: Line 403 and elsewhere.  Overall, it seems that the PPC/PSC ratios provide relatively little explanatory power.  Maybe this could be mentioned somewhere and the data not explicitly included, unless they provide additional useful information. In Fig. 7e, for example, all the points fall together.
**Response:** we think that the fact that these high-PPC points either stand out of the general relationship or, on the contrary and as noted here, fall together with the high-PSC points, show that pigment composition has an impact. These PPC-dominated data correspond to the bulk of surface (<50 m) oligotrophic waters, where the range of variability of IOPs and Chl is small.

**Reviewer comment or question**: Line 456: It's worth noting that the POC – Tchl data were distributed over a much smaller range.  If you took a similarly narrow range of other variables, relationships might also not be statistically significant.
**Response**: yes, we agree.

**Reviewer comment or question**: Line 466: worth emphasizing here the significant size bias of the cytometry data

**Response**: we think this point is addressed in the discussion and not really relevant here in the results section.

**Reviewer comment or question**: Fig. 6: One on hand, I can understand why the authors used log transformation for visual purposes, but the fact that this is not applied consistently across the panels makes direct comparison rather difficult. I would use all linear or log scaling.

**Response**: See our response above and also Fig. S2.

**Reviewer comment or question**: Line 485. I can see how the black points sit above the line, but I'm not sure if these data 'stand out' considering the magnitude of the error bars.

**Response:** we have modified the sentence as follows:

"The cluster of PPC-dominated data (black dots) does not overlap the general $b_{bp}(470)$-derived....."

**Reviewer comment or question**: Line 501: replace 'when' at the end, by 'while'

**Response:** the sentence has been further modified following a comment by the other reviewer.

**Reviewer comment or question**: Line 505: replace 'on' with 'in'

**Response:** done

**Reviewer comment or question**: Line 509: I can't really see the green line.

**Response**: we have made it thicker, so it is hopefully more visible now (same for the dashed blue line)

**Reviewer comment or question**: Fig. 7: I think it would be useful to plot the various Cphyto estimates against each other.

**Response**: that is actually what Fig. 8 does.

**Reviewer comment or question**: Line 520: the PPC-dominated points are distributed over a very narrow range, which could explain the lack of a clear relationship. Are they 'separated' beyond the error bars of measurements?

**Response:** yes they are.

**Reviewer comment or question**: Line 573: maybe 'refractive' instead of 'refringent' (I had to look that up).

**Response:** we think refringent is the right word here, as a property of cells. Refringent meaning "generating refraction".

**Reviewer comment or question**: Line 617. In addition to PSC/PPC ratios, there are other approaches to pigment-based taxonomy, including Chemtax, for example. Would different results have been achieved with another method? At the least, I think it would be useful to provide more information on the taxonomic composition of PPC-dominated waters. This comes up again on line 646.

**Response:** we actually looked at various pigment assemblages and also to the pico-, nano- and micro-phytoplankton relative contributions using the HPLC data and did not find any clear pattern connected to $C_{phyto}$ and IOPs variability.

This is now said in that paragraph.

**Reviewer comment or question**: Line 624: insert reference after 'ratios'

**Response:**

Gibb, S.W., Barlow, R.G., Cummings, D.G., Rees, N.W., Trees, C.C., Holligan, P., Suggett, D., 2000. Surface phytoplankton pigment distributions in the Atlantic Ocean: an assessment of basin scale variability between 50°N and 50°S. Prog. Oceanogr. 45, 339–368.

**Reviewer comment or question**: Line 626: insert reference under 'conditions'

**Response:**

Araujo, Milton Luiz Vieira, et al. "Contrasting patterns of phytoplankton pigments and chemotaxonomic groups along 30 S in the subtropical South Atlantic Ocean." Deep Sea Research Part I: Oceanographic Research Papers 120 (2017): 112-121.

**Reviewer comment or question**: Line 643: is the higher bbp due to larger surface area / volume ratio?

**Response:** it might be. Although we do not really have elements to support this we have nevertheless added it in the text.

**Reviewer comment or question**: Line 690: There is no doubt that the cytometer provides a significantly size-biased view of the community. I wonder how robust the ffc factor is across the different assemblages.

**Response:** we have rewritten the sentence as follows, because we do not need to question "whether" the cytometry technique misses a "significant" part of the population but how much it misses:

"Therefore, the question arises as to how much of the phytoplankton population the cytometry technique misses"

END OF REVIEW

---

## Author Comment (AC4)

[revised manuscript text omitted]

**2.4. Inherent optical properties (IOPs)**

A 25-cm pathlength Western Environmental Technology Laboratories (WET Labs) Inc. (Philomath, Oregon) C-star transmissometer was connected to the auxiliary channels of the CTD to measure the transmittance (Tr, %) at 660 nm at each cast. Prior to CTD casts, instrument windows were cleaned with tissue paper and isopropyl alcohol. The absorption of colored dissolved organic matter at 660 nm is assumed to be negligible, so the particulate beam attenuation coefficient, $c_p$, is obtained as follows:

$$c_p(660) = \frac{-\ln(Tr)}{0.25}, \qquad [m^{-1}] \tag{1}$$

where Tr is the transmission corrected for the water contribution and 0.25 the path length in meter.

A Hydro-Optics, Biology and Instrumentation Laboratories (HOBILabs Inc., San Diego, California) Hydroscat-6 (Maffione & Dana, 1997) was deployed to measure the total volume scattering function, $\beta$, ($m^{-1}$ $sr^{-1}$) at 140 degrees and six wavelengths (420, 442, 470, 510, 590, 700 nm). This instrument was part of an optical package deployed at each station immediately after the CTD cast. The sensor was factory calibrated before and after the cruise and dark cast measurements were performed in situ systematically before each cast, using black electric tape covering the instrument windows. The deployment included a rinse and temperature equilibration cast down to 50 m and back to the surface, after which the full cast started down to ~ 200 m at a descending speed of ~0.2 m $s^{-1}$. This speed, combined with an acquisition frequency of 1 Hz, resulted in a vertical resolution of about 0.25 m. 
[revised manuscript text omitted]